# ScenePilot: Controllable Boundary-Driven Critical Scenario Generation for Autonomous Driving

Qiyu Ruan [* 1 2]   Yuxuan Wang [* 1]   He Li [1 2]   Zhenning Li [1]   Chengzhong Xu [1 2]

## Abstract

Safety-critical scenarios are central to evaluating autonomous driving systems, yet their rarity in naturalistic logs makes simulation-based stress testing indispensable. Most scenario generation methods treat surrounding agents as adversaries, but they either (i) induce failures without explicitly modeling vehicle-road physical limits, yielding visually extreme yet physically unsolvable crashes, or (ii) enforce physical feasibility or policy feasibility in isolation, which can over-focus on aggressive maneuvers or remain tied to a controller-dependent capability boundary. We propose ScenePilot, a feasibility-guided, boundary-driven framework that targets the boundary band: scenarios that are physically solvable in principle yet still cause the deployed autonomy stack to fail. We formulate generation as constrained multi-objective reinforcement learning, combining an RSS-derived physical-feasibility score $\sigma$ with an online-learned AV-risk predictor $\Phi$, and introduce step-level feasibility-aware shielding to keep exploration near the feasibility boundary while avoiding infeasible artifacts. Experiments on SafeBench with multiple planners show that ScenePilot yields substantially higher collision rates (+6.2 percentage points) while preserving physical validity, and that adversarial fine-tuning on these boundary-band scenarios consistently reduces downstream crash rates. The code is available at https://github.com/QiyuRuan/ScenePilot.

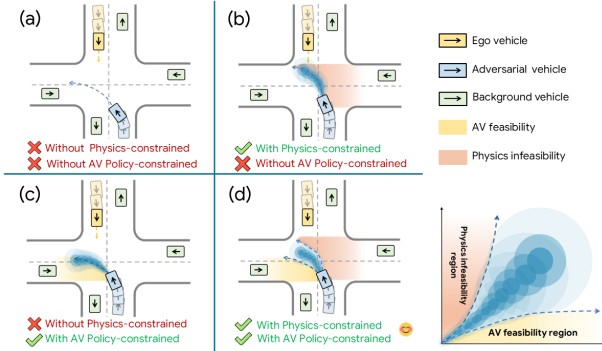

*Figure 1.* Illustration of four interaction regimes relative to AV controller and physical feasibility.

## 1. Introduction

Safety-critical scenarios are rare in real traffic but decisive for autonomous vehicles (AVs). Large-scale naturalistic driving logs cover everyday interactions and support AV behavior modeling (Liao et al., 2025; 2024; Wang et al., 2025), yet truly high-consequence events occupy only a tiny fraction of the data (Fig. 1(a)) (Liu & Feng, 2024). As a result, AV stacks that appear reliable under passive replay may still fail on rare but consequential edge cases. Although traffic-rule compliance, online verification, intelligent-transportation studies, and crash-severity analysis can support AV safety from runtime-assurance or risk-analysis perspectives (Althoff et al., 2025; Althoff & Dolan, 2014; Li et al., 2024a;b), they are not active tools for discovering diverse failure-inducing interactions. This motivates simulation-based scenario generation as a complementary evaluation tool for systematically exposing rare and informative safety-critical scenarios (Sun et al., 2021).

In simulation, critical scenario generation is commonly posed as an adversarial problem: surrounding vehicles are treated as agents that perturb the ego vehicle toward failure and are trained with reinforcement learning or generative models to induce collisions beyond naturalistic replay frequency (Wachi, 2019). While effective at increasing failure counts, many approaches implicitly optimize a single goal—*make the ego crash*—without controlling whether the resulting failures are informative or even physically

---

[*]Equal contribution  [1]State Key Laboratory of Internet of Things for Smart City (SKL-IOTSC), University of Macau, Macau, China  [2]Department of Computer and Information Science, University of Macau, Macau, China. Correspondence to: Chengzhong Xu <czxu@um.edu.mo>, Zhenning Li <zhenningli@um.edu.mo>.

*Proceedings of the 43rd International Conference on Machine Learning*, Seoul, South Korea. PMLR 306, 2026. Copyright 2026 by the author(s).

meaningful (Ding et al., 2020; Lu et al., 2022; Niu et al., 2023). In particular, when vehicle–road physical limits are not modeled explicitly, adversaries may exploit unrealistic accelerations, timing, or geometry to create visually aggressive but physically unsalvageable crashes, where no admissible control could have avoided the collision (Ghodsi et al., 2021; Rempe et al., 2022). These cases stress numerical robustness, but they conflate *difficulty* with *physical invalidity*, making it unclear whether failures reveal genuine deficiencies of the autonomy stack.

A natural response is to incorporate *physical feasibility* into scenario generation. Physics-constrained methods restrict adversarial perturbations to physically realizable behaviors. AdvSim (Wang et al., 2021) perturbs agents' trajectories while maintaining physically plausible motion and consistent sensor observations, and ACERO (Song et al., 2023) searches for realistic, executable maneuvers that can reliably trigger failures in closed-loop simulation. These approaches rule out obviously unrealistic interactions and produce plausible stress tests (Fig. 1(b)). However, using physical feasibility as the primary guiding constraint can cause generation to collapse onto extreme boundary-pushing maneuvers near the physical limits, yielding a skewed set of highly adversarial samples that under-represent more typical near-boundary interactions (Stoler et al., 2025). More broadly, heavy reliance on such highly adversarial samples can harm nominal performance and does not necessarily translate into actionable insights for improving the deployed autonomy stack (Raghunathan* et al., 2019).

Complementary to physics-constrained generation, another line of work constrains adversariality by the *AV policy capability* (Fig. 1(c)). FREA (Chen et al., 2025), for example, leverages a policy-feasibility signal to steer generation toward failures near the stack's capability boundary. This controller-aware view can be actionable for diagnosing a specific stack, but it anchors generation to a controller-dependent boundary and does not reveal how close the AV operates to the physical limits of the environment. In particular, states deemed unrecoverable by the deployed stack may still be physically solvable under admissible dynamics, so policy-constrained generation can miss the most informative regime where failures are *avoidable in principle* yet persist due to stack limitations.

This paper adopts a simple organizing view: scenario difficulty is governed by two factors—**physical feasibility** of vehicle–road interaction and **AV policy capability** of the deployed autonomy stack. Their interplay defines four regimes (Fig. 1): (a) nominal interactions far from both limits; (b) physically feasible but not necessarily informative adversarial cases produced by physics-constrained generation; (c) controller-bounded cases produced by policy-constrained generation; and (d) a particularly informative boundary band

where interactions remain physically solvable, yet still break the deployed stack. We argue that this boundary band best isolates competence gaps: failures that are not inevitable from physics, but arise from limitations of the current autonomy stack.

Building on this view, we propose **ScenePilot**, a feasibility-guided adversarial scenario generation framework that targets the boundary band (Fig. 1(d)). ScenePilot explicitly separates the two factors by quantifying physical feasibility using an RSS-derived score $\sigma$ and estimating stack vulnerability with an online-learned risk predictor $\Phi$. We formulate adversarial scenario generation as a constrained multi-objective Markov decision process and introduce *step-level feasibility-aware shielding* that prioritizes feasibility recovery when the adversary approaches infeasible interactions, while maintaining pressure toward high-risk failures. To systematically explore the near-boundary regime rather than collapsing onto a narrow set of extreme maneuvers, we further employ *feasibility-threshold sweeping* with $\varepsilon$, which controls how close generation operates to the physical feasibility boundary and concentrates samples in the physically feasible yet policy-infeasible band.

In summary, our contributions are as follows:

- We propose a feasibility-guided formulation that disentangles physical feasibility from AV policy capability and targets their boundary band, where scenarios are physically solvable but still induce failures of the deployed autonomy stack.

- We develop a constrained multi-objective adversarial generator that couples physical and policy signals $(\sigma, \phi)$ with step-level feasibility-aware shielding and feasibility-threshold sweeping to concentrate on physically feasible yet policy-infeasible near-boundary scenarios.

- On SafeBench with multiple planners and controllers, ScenePilot generates more safety-critical scenarios while keeping infeasible interactions rare, and adversarial fine-tuning on boundary-band scenarios consistently reduces downstream crash rates.

## 2. Related Work

### 2.1. Scenario-based Safety Testing and Generation

Scenario-based testing is a key approach for assessing AV safety (Neurohr et al., 2020). Existing methods for constructing safety-critical scenarios are often grouped into three families: data-driven, adversarial, and knowledge-based generation (Ding et al., 2023). Data-driven methods replay or perturb real trajectories in large naturalistic datasets or simulators (Li et al., 2019; Yang et al., 2023; Feng et al., 2021), and

thus offer strong realism but are constrained by the long-tail rarity and collection cost of critical events (Lu et al., 2026). Adversarial methods directly manipulate surrounding vehicles to induce near-collisions or collisions (Ding et al., 2021; Jia et al., 2024), improving efficiency over passive logs but often optimizing failure frequency without strictly enforcing vehicle-road physics (Chen et al., 2021). Knowledge-based methods encode traffic rules and expert priors into scenario templates or constraint systems (Ding et al., 2025), improving controllability but relying on hand-crafted coverage. Overall, prior work emphasizes realism, adversariality, or rule coverage, with some methods incorporating physical feasibility constraints (Dong et al., 2025). However, existing methods often skew toward either unconstrained failure maximization or heavily physics-focused boundary cases, underexploring the near-boundary band of physically solvable yet AV policy-breaking interactions. We complement these lines by explicitly adopting a boundary-focused view and targeting this near-boundary feasible band.

## 2.2. RL for Scenario Generation

Reinforcement learning (RL) is a natural tool for scenario generation, as AVs and surrounding agents interact in a sequential decision process. RL-based methods train adversarial agents to control nearby vehicles or modify traffic configurations and thereby synthesize safety-critical encounters (Liu et al., 2024; Wei et al., 2024). Many objectives directly encourage collisions or near-misses, which is effective for exposing failures quickly but may also drive adversaries toward physically implausible or trivially infeasible behaviors, reducing the diagnostic value of the resulting scenarios (Kuutti et al., 2020). To improve physical validity, some works (Hao et al., 2023; Cai et al., 2024) incorporate physics-aware constraints modeling into RL. However, such designs can bias generation toward extreme, boundary-hugging feasible interactions, rather than the broader near-boundary band that remains physically solvable yet exposes algorithmic weaknesses. Chen et al. (Chen et al., 2025) further propose a boundary-aware formulation by defining the Largest Feasible Region (LFR) for a given AV controller and shaping adversaries toward its boundary. Yet LFR is controller-dependent, mixing controller recoverability with physical feasibility. We instead target a controller-independent physical feasibility boundary and focus on physically solvable but AV policy-breaking scenarios in the near-boundary band.

## 3. Methods

ScenePilot is an adversarial scenario generation framework (Fig. 2), where a learnable scenario policy controls non-ego agents to evaluate ego autonomous driving policy under challenging interactions. We characterize each rollout with two per-step signals: an AV-risk and a physics-based feasibility.

The scenario policy is trained to concentrate rollouts in a boundary band that is physically feasible yet beyond the ego controller's capability (AV policy-infeasible), rather than producing trivial cases or physically impossible collisions.

### 3.1. Problem Formulation

**Adversarial MDP.** We formulate critical scenario generation as an adversarial discounted multi-objective MDP $\langle \mathcal{S}, \mathcal{A}, P, \mathbf{r}, \gamma \rangle$, where a learnable scenario policy $\pi(\cdot \mid s_t)$ controls all non-ego agents. At time $t$, the driving environment is in state $s_t \in \mathcal{S}$ and the scenario policy samples $a_t \sim \pi(\cdot \mid s_t)$, where $a_t$ denotes the joint action of the non-ego agent set. The ego vehicle follows a fixed driving policy $\pi^{\text{ego}}$ and is absorbed into the environment dynamics, so that transitions satisfy $s_{t+1} \sim P(\cdot \mid s_t, a_t)$. Each transition is annotated with a two-dimensional reward vector signal $\mathbf{r}_t = (\phi_t, \sigma_t)$.

**Two-signal objective and boundary band.** The two components of $\mathbf{r}_t$ play complementary roles. The AV-risk signal $\phi_t$ measures how close the ego is to failure (larger is riskier), while $\sigma_t$ is a physics-based feasibility derived from kinematic limits. Specifically, $\sigma_t \geq 0$ indicates physically feasible frames, and $\sigma_t < 0$ indicates physically infeasible (i.e., unavoidable-collision) frames. We define discounted objectives $J_\phi(\pi) = \mathbb{E}_\pi \left[ \sum_{t=0}^{T-1} \gamma^t \phi_t \right]$ and $J_\sigma(\pi) = \mathbb{E}_\pi \left[ \sum_{t=0}^{T-1} \gamma^t \sigma_t \right]$. Our goal is to generate scenarios that are physically feasible yet AV policy-infeasible. We therefore focus on the boundary band induced in the $(J_\phi, J_\sigma)$ plane, $\mathcal{B} \triangleq \{\pi \mid J_\sigma(\pi) \geq 0, \; J_\phi(\pi) \geq \delta\}$, where $J_\sigma(\pi) \geq 0$ enforces physical feasibility and $J_\phi(\pi) \geq \delta$ filters trivial low-risk cases. To traverse this band and obtain a spectrum of physically feasible yet AV policy-infeasible scenarios, we optimize a family of $\varepsilon$-constrained objectives by Eq. (1), where $\varepsilon$ is a control variable to select different feasibility levels during training.

$$\max_\pi \; J_\phi(\pi) \qquad \text{s.t.} \qquad J_\sigma(\pi) \geq \varepsilon. \qquad (1)$$

### 3.2. Safety Signals

**Physical Safety.** We develop our notion of physical safety based on the Mobileye Responsibility-Sensitive Safety (RSS) model (Shalev-Shwartz et al., 2017), a widely adopted and principled framework for formalizing safety in autonomous driving. RSS encodes "duty-of-care" driving rules and provides analytically checkable safe-distance conditions in both longitudinal and lateral dimensions. If all vehicles comply with these rules, traffic should be accident-free. As such, RSS provides a principled baseline for distinguishing physically feasible yet unsafe interactions from those consistent with safe driving. However, the RSS safe-distance formulas are intentionally conservative (Hassanin et al., 2022).

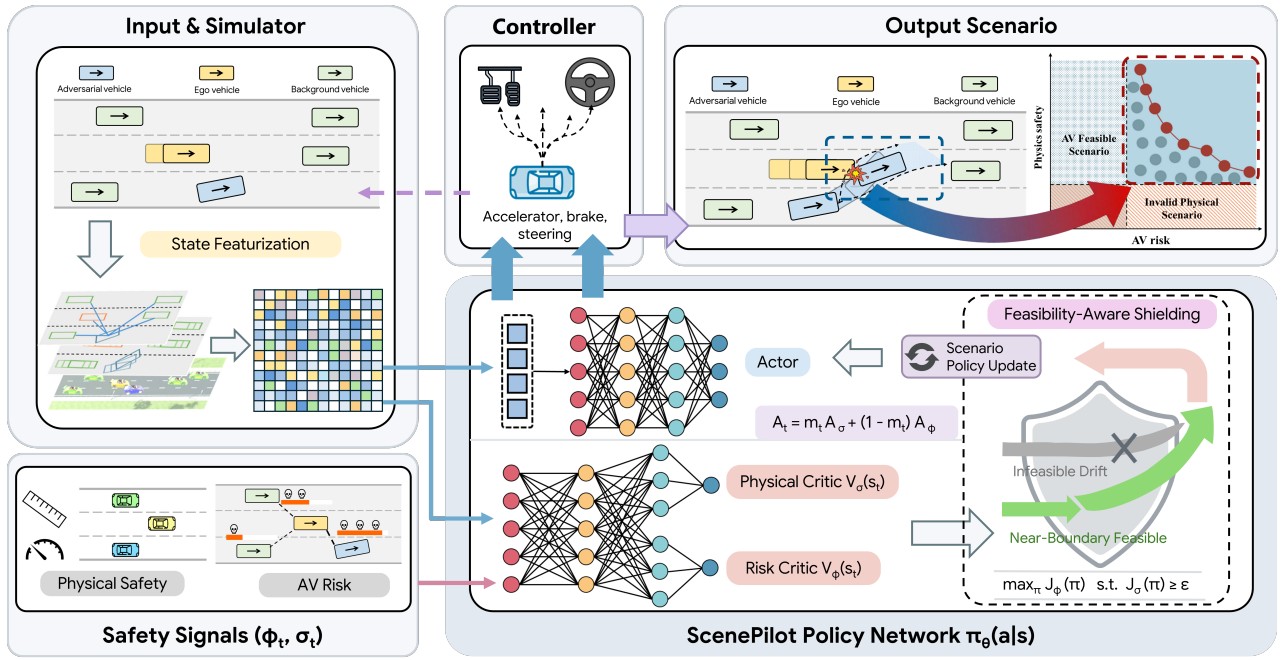

*Figure 2.* Overview of our ScenePilot framework. We characterize each rollout with an AV risk signal and a physics feasibility signal, and train a scenario policy to produce scenarios concentrated on the physically feasible yet AV policy-infeasible boundary band.

Such conservatism is desirable for deployment-level safety guarantees, but it is counterproductive for critical scenario exploration. Many near-crash yet still physically avoidable frames would be flagged as unsafe and thus prematurely excluded. To better characterize the theoretically solvable region, we derive a physics-limit variant of the RSS longitudinal and lateral distances in Eq. (2).

$$d_{\text{opp}}^{\text{lon}} = \frac{(v_r^{\text{lon}})^2}{2a_{r,\text{max}}^{\text{lon,brake}}} + \frac{(v_f^{\text{lon}})^2}{2a_{f,\text{max}}^{\text{lon,brake}}}, \tag{2a}$$

$$d_{\text{same}}^{\text{lon}} = \max\left[ \underbrace{\frac{(v_r^{\text{lon}})^2}{2a_{r,\text{max}}^{\text{lon,brake}}} - \frac{(v_f^{\text{lon}})^2}{2a_{f,\text{max}}^{\text{lon,brake}}}}_{\text{stop}}, \underbrace{\frac{(v_r^{\text{lon}} - v_f^{\text{lon}})^2}{2\left(a_{r,\text{max}}^{\text{lon,brake}} - a_{f,\text{max}}^{\text{lon,brake}}\right)}}_{\text{equal}} \right], \tag{2b}$$

$$d_{\text{same}}^{\text{lat}} = \frac{(v_r^{\text{lat}})^2}{2a_{r,\text{max}}^{\text{lat,brake}}} - \frac{(v_f^{\text{lat}})^2}{2a_{f,\text{max}}^{\text{lat,brake}}}, \tag{2c}$$

$$d_{\text{opp}}^{\text{lat}} = \frac{(v_r^{\text{lat}})^2}{2a_{r,\text{max}}^{\text{lat,brake}}} + \frac{(v_f^{\text{lat}})^2}{2a_{f,\text{max}}^{\text{lat,brake}}}. \tag{2d}$$

We index the two interacting vehicles by $x \in \{r, f\}$, where $r$ is the rear (left in lateral) vehicle and $f$ is the front (right in lateral) one. The scalars $v_x^{\text{lon}}$ and $v_x^{\text{lat}}$ denote the longitudinal and lateral speed components of vehicle $x$, and $a_{x,\text{max}}^{\text{lon,brake}}$

and $a_{x,\text{max}}^{\text{lat,brake}}$ denote its maximal feasible decelerations along the corresponding axis. This variant corresponds to an idealized limit in which vehicles react instantaneously and decelerate at their maximal feasible rates under kinematic constraints, thereby providing an axis-wise braking-limit distance $d_{\text{limit}}$. Consequently, if the actual edge-to-edge clearance on an axis satisfies $(|d_{\text{actual}}| - s(\Delta\psi)) < d_{\text{limit}}$ (Eq. (2)), then a collision is unavoidable under braking-only responses along that axis.

$$\boldsymbol{d}_{\text{actual}} = \begin{bmatrix} d_{\text{actual},x} \\ d_{\text{actual},y} \end{bmatrix} = \begin{bmatrix} \cos\psi_{\text{ego}} & \sin\psi_{\text{ego}} \\ -\sin\psi_{\text{ego}} & \cos\psi_{\text{ego}} \end{bmatrix} \left( \begin{bmatrix} x_{\text{adv}} \\ y_{\text{adv}} \end{bmatrix} - \begin{bmatrix} x_{\text{ego}} \\ y_{\text{ego}} \end{bmatrix} \right),$$

$$\boldsymbol{s}(\Delta\psi) = \begin{bmatrix} s_x \\ s_y \end{bmatrix} = \begin{bmatrix} L_{\text{ego}} + |\cos\Delta\psi| L_{\text{adv}} + |\sin\Delta\psi| W_{\text{adv}} \\ W_{\text{ego}} + |\cos\Delta\psi| W_{\text{adv}} + |\sin\Delta\psi| L_{\text{adv}} \end{bmatrix},$$

$$\sigma = 1 - \left\| \left( \boldsymbol{d}_{\text{limit}} - (|\boldsymbol{d}_{\text{actual}}| - \boldsymbol{s}(\Delta\psi)) - \boldsymbol{l} \right)_+ \oslash \boldsymbol{d}_{\text{limit}} \right\|_p. \tag{3}$$

Importantly, braking-only infeasibility along the imminent-collision axis does not preclude collision avoidance under coupled braking-steering maneuvers, as the orthogonal axis may provide sufficient reachable displacement within the available time-to-collision. To quantify instantaneous physical feasibility beyond braking, we decompose each frame in the ego-fixed body frame into longitudinal $(x)$ and lateral $(y)$ axes and evaluate the residual margins on both. We then incorporate time-to-collision and the reachable displacement under bounded accelerations to aggregate these margins into a unified physical safety score $\sigma$ in Eq. (3).

$\boldsymbol{d}_{\text{actual}}$ denotes the ego-adversarial vehicle center distance

expressed in the ego body frame, $s(\Delta\psi)$ denotes the projected envelope of the two oriented bounding boxes under the relative yaw between the ego and adversarial vehicles, $\Delta\psi = \psi_{\text{adv}} - \psi_{\text{ego}}$, and $d_{\text{limit}}$ denotes the physics-limit safe distance from Eq. (2) projected onto the longitudinal and lateral axes. The operator $(\cdot)_+$ acts elementwise as $[z]_+ = \max(0, z)$, $\oslash$ denotes elementwise division, and $\|\cdot\|_p$ is the $\ell_p$ norm, which aggregates the normalized residual gaps on the longitudinal and lateral axes. The vector $\boldsymbol{l} = [l_x, l_y]^\top$ encodes the reachable displacements that can be compensated along the axis orthogonal to the imminent collision, with $t$ determined by Eq. (4).

$$t = \min\left(\underbrace{\frac{[|d_x^{\text{actual}}| - s_x(\Delta\psi)]_+}{[\Delta v_x]_+}}_{\text{TTC (lon)}}, \underbrace{\frac{[|d_y^{\text{actual}}| - s_y(\Delta\psi)]_+}{[\Delta v_y]_+}}_{\text{TTC (lat)}}\right) \quad (4)$$

Specifically, we compute the axial time to collision $(t_x, t_y)$ and identify the first-colliding axis $c = \arg\min\{t_x, t_y\}$ and its orthogonal axis $n = \{x, y\} \setminus \{c\}$. The displacement along the colliding axis is set to zero ($l_c = 0$), while that on the orthogonal axis is compensated as $l_n = \frac{1}{2} a_n^{\text{rel}} t^2 \mathbf{1}\{\Delta v_n > 0\}$, where $\mathbf{1}\{\cdot\}$ is the indicator function, $a_n^{\text{rel}}$ is the maximal feasible relative acceleration and $\Delta v_n > 0$ denotes a closing motion. By construction, $\sigma \geq 0$ indicates physically feasible frames, and $\sigma < 0$ indicates physically infeasible frames where no admissible combination of braking and steering can prevent a collision. Details are provided in Appendix A.1.

**AV Risk.** FREA's feasibility-aware risk (LFR) (Chen et al., 2025) offers a certificate indicating whether a state lies within the AV's feasible set, making it well-suited for characterizing the trajectory-level feasibility boundary. However, this certificate and its explicit constraint modeling are unnecessary in our work. We therefore introduce a lightweight, online-learnable AV-risk network that estimates the ego vehicle's instantaneous safety risk $\phi$ at each frame. Specifically, we learn this predictor as a policy-free critic (value learning without an actor). Given a scenario frame feature $s_t$, we regress the network output $\hat{\Phi}(s_t) \in [0, 1]$ to a potential-shaped temporal-difference target:

$$y_t = \underbrace{\mathbb{1}\{\mathcal{C}_t\}}_{\text{collision}} + \gamma\,\hat{\Phi}(s_{t+1}) + \underbrace{\gamma\,F(s_{t+1}) - F(s_t)}_{\text{potential shaping}} \quad (5)$$

where $\mathcal{C}_t$ denotes the collision event at time $t$. The potential-based shaping term densifies the otherwise sparse collision feedback and accelerates learning, while preserving optimal policies (Ng et al., 1999). In our implementation, we choose $F(s) = \kappa/d(s)$, where $d(s)$ denotes the ego–nearest-adversary distance and $\kappa > 0$ is a scaling factor.

The network is a two-layer MLP with sigmoid output and is trained online by a weighted binary cross-entropy to handle class imbalance in Eq. (6), where a larger weight $w_+ \gg 1$ emphasizes rare collision targets.

$$\mathcal{L}_t = w_t \cdot \text{BCE}\big(\hat{\Phi}(s_t), \text{clip}(y_t, 0, 1)\big),$$
$$w_t = \begin{cases} w_+, & \mathcal{C}_t \text{ is true,} \\ 1, & \text{otherwise.} \end{cases} \quad (6)$$

Under potential-based shaping, the learned value is a shifted estimate, $\hat{\Phi}(s) = \Phi(s) - F(s)$. At inference, we recover the unshifted AV risk by adding back the potential, $\Phi(s) = \text{clip}\big(\hat{\Phi}(s) + F(s), 0, 1\big)$. This yields a continuous, differentiable score that provides an interpretable measure of the ego vehicle's risk across states. Details are provided in Appendix A.2.

### 3.3. ScenePilot

Given per-frame AV-risk and physical-safety reward signals, our goal is to solve the $\varepsilon$-constrained objective in Eq. (1) and generate scenarios that are risky for AV yet physically solvable. We use the two components of the per-step reward vector $\mathbf{r}_t = (\phi_t, \sigma_t)$ and sweep the feasibility threshold $\varepsilon$ during training to traverse a near-boundary feasible band. Inspired by CMORL-IPO (Liu et al., 2025), a natural way to enforce $J_\sigma \geq \varepsilon$ is an interior-point scalarization, i.e., optimizing $J_\phi + \lambda J_\sigma$ with $\lambda \propto (\mathbb{E}[J_\sigma] - \varepsilon)^{-1}$. While effective for exploring global trade-offs, this batch-mean construction can smooth out rare near-boundary frames and weaken responsiveness to imminent violations. We therefore adopt a step-level shielding mechanism and learn dual value critics, following safe-RL formulations (Wagener et al., 2021; Yu et al., 2022). We use a deterministic state featurization that concatenates privileged simulator signals, including ego state, surrounding agents, route cues, and map features, into a vector state $s_t$, which conditions the actor and a two-headed vector-valued critic.

$$\mathbf{V}(s_t) = \mathbb{E}\Big[\sum_{k=0}^{T-1} \gamma^k \mathbf{r}_{t+k} \,\Big|\, s_t\Big], \quad \mathbf{r}_t = (\phi_t, \sigma_t). \quad (7)$$

From rollouts, we form the vector return $\mathbf{G}_t = \sum_{k=0}^{T-1} \gamma^k \mathbf{r}_{t+k}$ and train the two-head critic by a vector regression loss in Eq. (8), where $\mathbf{V}(s_t) = (V_\phi(s_t), V_\sigma(s_t))$.

$$\mathcal{L}_V = \mathbb{E}\Big[\big\|\mathbf{V}(s_t) - \mathbf{G}_t\big\|_2^2\Big]. \quad (8)$$

We then compute generalized advantage estimates (GAE) for both components using the corresponding TD residuals. Specifically, let $\boldsymbol{\delta}_t = \mathbf{r}_t + \gamma\mathbf{V}(s_{t+1}) - \mathbf{V}(s_t)$. We apply $\text{GAE}(\gamma, \lambda_{\text{gae}})$ to each component of $\boldsymbol{\delta}_t$ to obtain $\mathbf{A}_t = (A_t^\phi, A_t^\sigma)$. To retain sensitivity to imminent feasibility violations, we introduce a step-level shielding mask $m_t \triangleq \mathbb{I}[h_t > 0 \vee h_{t+1} > 0]$, with violation residuals

$h_t = [\varepsilon_t - \sigma_t]_+$ and $h_{t+1} = [\varepsilon_t - \sigma_{t+1}]_+$. The threshold $\varepsilon_t$ follows the Gaussian-shaped schedule in Eq. (33) of Appendix A.3. The shielded advantage is

$$\tilde{A}_t = m_t A_t^\sigma + (1 - m_t) A_t^\phi. \qquad (9)$$

This implements a step-level shielding principle. Whenever a feasibility violation is present or imminent, policy updates prioritize feasibility recovery via $A_t^\sigma$; otherwise they optimize risk via $A_t^\phi$. In contrast, an episode- or batch-mean feasibility penalty can dilute these short but decisive near-boundary violations, allowing a few infeasible frames to be traded off against many feasible ones. With a diagonal-Gaussian actor $\pi_\theta(a_t \,|\, s_t)$, we plug the shielded advantage $\tilde{A}_t$ into a PPO-style clipped surrogate:

$$
\begin{aligned}
\mathcal{L}_{\text{policy}}(\theta) = & - \mathbb{E}_t \Big[ \min \Big( \rho_t(\theta) \, \tilde{A}_t, \ \bar{\rho}_t(\theta) \, \tilde{A}_t \Big) \Big] \\
& - c_{\text{ent}} \, \mathbb{E}_t \Big[ \mathcal{H} \big( \pi_\theta(\cdot \mid s_t) \big) \Big],
\end{aligned}
\qquad (10)
$$

where $\rho_t(\theta) = \exp\big( \log \pi_\theta(a_t \,|\, s_t) - \log \pi_{\theta_{\text{old}}}(a_t \,|\, s_t) \big)$ and $\bar{\rho}_t(\theta) = \min\{1 + \eta, \max\{1 - \eta, \rho_t(\theta)\}\}$. We use standard entropy regularization and gradient clipping, and train the critic with Eq. (8). The moving threshold $\varepsilon$ sweeps the feasible band over training, while step-level shielding in Eq. (9) preserves sensitivity to rare near-boundary events that episode/batch averages may wash out. Together, these designs concentrate the scenario policy on physically feasible yet AV policy-infeasible cases along the boundary band.

# 4. Experiments

In this section, we evaluate ScenePilot on SafeBench (Xu et al., 2022) with two complementary goals. First, we assess how effectively the generated scenarios expose safety-critical ego failures while remaining physically plausible under vehicle–road limits. Second, we test downstream utility by adversarially fine-tuning ego policies on the generated scenarios and evaluating robustness on held-out routes.

All experiments are conducted in CARLA (Dosovitskiy et al., 2017) under the SafeBench pipeline (Xu et al., 2022), where our method controls adversarial agents and the ego is driven by a learning-based policy. This enables like-for-like comparison under the official protocol, and we additionally report physical plausibility statistics based on our RSS-derived feasibility score.

Our main findings are: (1) Compared with competitive baselines, ScenePilot generates more challenging yet physically plausible scenarios, avoiding failure cases dominated by physically infeasible artifacts. (2) Although learned against a fixed surrogate ego policy, the generated failures transfer across heterogeneous controllers. (3) Fine-tuning on ScenePilot scenarios yields the strongest downstream robustness under an otherwise identical training pipeline. (4)

ScenePilot covers a broader portion of the physically feasible yet policy-infeasible boundary band, rather than collapsing to a narrow set of extreme cases.

## 4.1. Setup

**Scenarios.** We follow SafeBench and generate/evaluate adversarial interactions on eight canonical traffic scenarios, each instantiated on ten pre-defined routes that cover diverse map topologies and interaction patterns (Straight/Turning Obstacle, Lane Changing, Vehicle Passing, Red-light Running, Unprotected Left-turn, Right Turn, and Crossing Negotiation). Unless otherwise stated, all scenario-generation statistics are aggregated over all routes within each base scenario. For downstream ego-policy fine-tuning, we adopt a route-disjoint split within each base scenario: routes 1–8 are used to collect training scenes, while routes 9–10 are held out for evaluation. Implementation details of generated scenario sampling are provided in Appendix B.4.

**AD algorithms.** To assess transfer and controller-agnostic difficulty, we evaluate each selected scene with three representative deep RL controllers provided by SafeBench: Proximal Policy Optimization (PPO) (Schulman et al., 2017), an on-policy stochastic method; Soft Actor-Critic (SAC) (Haarnoja et al., 2018), an off-policy stochastic method; and Twin Delayed DDPG (TD3) (Fujimoto et al., 2018), an off-policy deterministic method. Following SafeBench, the ego observation is a four-dimensional vector capturing (i) distance to the next waypoint, (ii) longitudinal speed, (iii) angular speed, and (iv) a binary indicator of a front-facing vehicle. Using this interface preserves strict comparability with prior reports. Beyond this standard RL-controller evaluation, we further evaluate generated scenarios on a broader set of heterogeneous AV stacks, including CARLA Autopilot (Dosovitskiy et al., 2017) (Traffic-Manager-controlled policy), AIM-BEV (Hanselmann et al., 2022) (privileged-input end-to-end planner), TransFuser (Prakash et al., 2021) (raw-sensor end-to-end planner), BehaviorAgent (Dosovitskiy et al., 2017) (rule-based CARLA agent).

**Baselines.** We compare against seven SOTA scenario-generation baselines in SafeBench, spanning two common design philosophies: (i) adversary-driven approaches that learn perturbations of initial states or trajectories, including Learning-to-Collide (LC) (Ding et al., 2020) and AdvSim (AS) (Wang et al., 2021); (ii) rule/optimization-driven approaches that encode traffic priors or trajectory costs, including the CARLA Scenario Generator (CS) (CARLA Scenario Runner Contributors, 2019) and Adversarial Trajectory Optimization (AT) (Zhang et al., 2022). We also include ChatScene (Zhang et al., 2024), SCSG (Karacik et al., 2025), and SCENGE (Liu et al., 2026), which are strong recent baselines that generate scenarios via scene code. All meth-

*Table 1.* **Statistics of scenario generation on SafeBench base scenarios.** We report the collision rate (CR) and overall score (OS) to quantify the safety-criticality of the generated scenarios. For each algorithm, metrics are averaged over three ego controllers (SAC, PPO, TD3) and all scenes under the same base scenario; the rightmost column summarizes the mean performance across all base scenarios, with bold entries indicating the best result among the compared methods. LC (Learning-to-Collide), AS (AdvSim), CS (CARLA Scenario Generator), and AT (Adversarial Trajectory Optimization) follow the standard SafeBench setup, while ChatScene (Zhang et al., 2024), SCSG (Karacik et al., 2025), and SCENGE (Liu et al., 2026) denote recent safety-critical scenario generators. ↑ / ↓ denote that higher/lower values are preferable for the corresponding metric.

| Metric | Algo. | Base Traffic Scenarios | | | | | | | | Avg. |
| | | Straight Obstacle | Turning Obstacle | Lane Changing | Vehicle Passing | Red-light Running | Unprotected Left-turn | Right turn | Crossing Negotiation | |
|---|---|---|---|---|---|---|---|---|---|---|
| CR ↑ | LC | 0.30 | 0.09 | 0.87 | 0.83 | 0.71 | 0.69 | 0.59 | 0.58 | 0.584 |
| | AS | 0.51 | 0.33 | 0.86 | 0.87 | 0.57 | 0.70 | 0.29 | 0.57 | 0.586 |
| | CS | 0.45 | 0.61 | 0.89 | 0.87 | 0.63 | 0.69 | 0.68 | 0.60 | 0.676 |
| | AT | 0.50 | 0.31 | 0.78 | 0.82 | 0.71 | 0.68 | 0.59 | 0.62 | 0.627 |
| | ChatScene | 0.89 | 0.70 | 0.95 | **0.93** | 0.79 | 0.75 | 0.78 | **0.86** | 0.831 |
| | SCSG | 0.76 | **0.84** | 0.72 | 0.89 | 0.70 | 0.64 | 0.76 | 0.56 | 0.734 |
| | SCENGE | 0.86 | 0.77 | 0.84 | 0.90 | 0.82 | 0.75 | 0.76 | **0.86** | 0.820 |
| | ScenePilot | **0.90** | **0.84** | **0.99** | 0.89 | **0.93** | **0.89** | **0.91** | 0.79 | **0.893** |
| OS ↓ | LC | 0.761 | 0.830 | 0.505 | 0.507 | 0.601 | 0.615 | 0.548 | 0.588 | 0.619 |
| | AS | 0.673 | 0.707 | 0.507 | 0.490 | 0.675 | 0.607 | 0.705 | 0.593 | 0.620 |
| | CS | 0.698 | 0.567 | 0.489 | 0.490 | 0.641 | 0.613 | 0.505 | 0.579 | 0.573 |
| | AT | 0.668 | 0.714 | 0.538 | 0.505 | 0.607 | 0.620 | 0.545 | 0.569 | 0.596 |
| | ChatScene | **0.470** | 0.522 | **0.434** | **0.440** | 0.537 | 0.560 | 0.474 | **0.421** | 0.482 |
| | SCSG | 0.537 | **0.497** | 0.570 | 0.477 | 0.540 | 0.610 | 0.523 | 0.597 | 0.544 |
| | SCENGE | 0.503 | 0.526 | 0.504 | 0.457 | 0.507 | 0.519 | 0.498 | 0.477 | 0.499 |
| | ScenePilot | 0.505 | 0.504 | 0.458 | 0.471 | **0.488** | **0.507** | **0.399** | 0.478 | **0.476** |

ods are run in the same simulator with matched sampling budgets, identical initialization and termination conditions, and the same interaction interface.

**Metrics.** We report two primary metrics used in SafeBench: (i) **Collision Rate (CR)**, the proportion of runs that end in a collision with the ego (higher implies stronger safety-criticality). (ii) **Overall Score (OS)**, the official composite metric aggregating Safety, Functionality, and Etiquette indicators (lower implies harder scenes). The definitions and computation follow the established SafeBench and ChatScene to ensure strict comparability across methods.

### 4.2. Evaluation of the Generated Safety-Critical Scenarios

We first evaluate the safety-criticality of the generated scenes under the standard SafeBench protocol, where each method's generated scenarios are replayed with three RL-based ego controllers (SAC, PPO, TD3). Metrics are averaged over the three controllers and all scenes within the same base scenario. As shown in Table 1, ScenePilot achieves the strongest overall safety-criticality across the eight SafeBench base scenarios, with the highest mean CR (0.893) and the lowest mean OS (0.476). Relative to prior SafeBench baselines (LC/AS/CS/AT) and recent strong generators ChatScene, SCSG and SCENGE, ScenePilot improves CR on most scenarios while consistently lowering OS, indicating that the generated failures are not tailored

to a single ego policy but transfer across controllers. Qualitative examples are shown in Fig. 3, where the red vehicle denotes the ego AV, the green vehicle denotes the adversarial vehicle, and the right panel shows the ego vehicle's front-view observation. The generated scenario stays close to the feasibility boundary while inducing failures across ego controllers. We provide a component-wise breakdown of OS (Safety/Functionality/Etiquette) in Appendix C for further diagnosis.

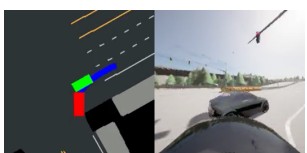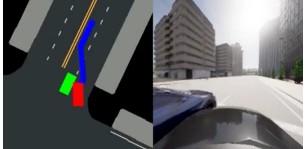

*Figure 3.* Visualization of a near-boundary scenario generated by ScenePilot.

To further examine whether ScenePilot remains effective across heterogeneous AV stacks, we conduct an additional study beyond the standard SafeBench RL-controller evaluation. We generate 100 SafeBench Scenario 6 cases using CARLA Autopilot as the ego stack, and replay the generated cases on Autopilot, AIM-BEV, TransFuser, Behavior-Agent, and TD3/SAC/PPO. We use KING (Hanselmann et al., 2022) as the baseline under the same Autopilot-based generation and SafeBench evaluation protocol. As shown in Table 2, ScenePilot remains more challenging than KING on average, increasing CR from 0.380 to 0.490 and reducing

*Table 2.* **Statistics of generated SafeBench Scenario 6 (Unprotected Left-turn) cases under heterogeneous AV stacks.** We report the collision rate (CR) and overall score (OS) to quantify the safety-criticality of generated scenarios under different ego stacks. The evaluated stacks include CARLA Autopilot (Dosovitskiy et al., 2017) (Traffic-Manager-controlled), AIM-BEV (Hanselmann et al., 2022) (privileged-input end-to-end), TransFuser (Prakash et al., 2021) (raw-sensor end-to-end), BehaviorAgent (Dosovitskiy et al., 2017) (rule-based), and TD3/SAC/PPO (deep-RL). ↑ / ↓ denote that higher/lower values are preferable for the corresponding metric.

| Metric | Algo. | Autopilot | AIM-BEV | TransFuser | BehaviorAgent | TD3 | SAC | PPO | Avg. |
|---|---|---|---|---|---|---|---|---|---|
| CR ↑ | KING | 0.09 | 0.00 | 0.04 | 0.77 | 0.20 | 0.60 | **0.96** | 0.380 |
| | ScenePilot | **0.11** | **0.15** | **0.16** | **0.77** | **0.57** | **0.79** | 0.88 | **0.490** |
| OS ↓ | KING | 0.909 | 0.953 | 0.939 | 0.577 | 0.858 | 0.654 | **0.475** | 0.766 |
| | ScenePilot | **0.895** | **0.871** | **0.879** | **0.570** | **0.665** | **0.560** | 0.516 | **0.708** |

OS from 0.766 to 0.708. These results indicate that ScenePilot is not limited to the standard RL-controller setting, but remains effective for more capable AV stacks.

## 4.3. Evaluation under Denser Traffic Conditions

*Table 3.* **Statistics of generated scenarios under denser background traffic.** We report the collision rate (CR) and overall score (OS) after inserting different numbers of Traffic-Manager-controlled background vehicles (BV). The primary adversarial vehicle remains optimized by ScenePilot. ↑ / ↓ denote that higher/lower values are preferable for the corresponding metric.

| Metric | BV | SAC | PPO | TD3 | Avg. |
|---|---|---|---|---|---|
| CR ↑ | 0 | **0.99** | 0.83 | 0.85 | 0.89 |
| | 30 | **0.99** | 0.84 | **0.89** | **0.91** |
| | 50 | **0.99** | **0.85** | 0.88 | **0.91** |
| OS ↓ | 0 | 0.457 | 0.536 | 0.528 | 0.507 |
| | 30 | 0.450 | 0.533 | **0.500** | 0.494 |
| | 50 | **0.446** | **0.524** | 0.505 | **0.492** |

We also evaluate ScenePilot under denser background traffic. Specifically, we insert 30 and 50 additional CARLA Traffic-Manager-controlled background vehicles into the SafeBench Scenario 6 (Unprotected Left-turn), while keeping the primary adversarial vehicle optimized by ScenePilot unchanged from standard SafeBench-generated ScenePilot scenarios. As shown in Table 3, ScenePilot remains effective under denser traffic conditions. Compared with the original setting without additional background vehicles, the average CR slightly increases from 0.89 to 0.91, while the average OS decreases from 0.507 to 0.492. These results suggest that denser surrounding traffic affects the realized adversarial interaction through occupancy, gap availability, and timing, making the scenarios slightly more difficult. However, since the added background vehicles are benign rather than adversarially optimized, the main stress-testing pressure still comes from ScenePilot's primary adversarial vehicle. Thus, ScenePilot preserves its safety-critical effect when the primary adversarial interaction is embedded in denser traffic.

## 4.4. Effectiveness of Downstream Utility on Fine-tuning

We further evaluate whether the generated scenes provide actionable learning signals for improving an ego policy via adversarial fine-tuning. Following ChatScene (Zhang et al., 2024), we fine-tune the same surrogate SAC ego policy used for scenario generation, ensuring an apples-to-apples comparison. For each base scenario, we fine-tune the ego separately on the safety-critical scenes generated from the first eight routes, and evaluate on a held-out set from the remaining two routes with test scenes pooled across generators (about 40 cases per scenario; details in Appendix B.5). Table 4 shows that fine-tuning on generated critical scenarios substantially improves robustness over the pre-trained surrogate (PP). On average, CR decreases from 0.854 to 0.134/0.072 after fine-tuning on ChatScene/ScenePilot scenes, while OS increases from 0.525 to 0.869/0.898. ScenePilot achieves the best post-finetuning robustness with the lowest average CR (0.072) and the highest OS (0.898). Overall, these results suggest that ScenePilot produces training scenes with stronger downstream utility, leading to better generalization under the same fine-tuning pipeline.

## 4.5. AV–Physics Analysis of Generated Critical Scenarios

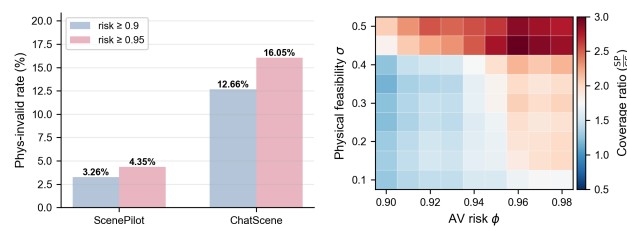

*(a)* Phys-invalid frame rate.    *(b)* Risk-physics coverage ratio.

*Figure 4.* **Quantitative characterization of the AV–physics gap between ScenePilot and ChatScene.** (a) Physically invalid frame rate under different AV-risk thresholds. (b) Coverage ratio of ScenePilot to ChatScene in the AV-risk–physical-feasibility space.

To better understand the generated scenarios, we analyze their AV–physics characteristics. Fig. 4(a) compares the

*Table 4.* **Evaluating Ego-Vehicle Performance Post-Finetuning.** This table evaluates how effectively each method's generated scenarios serve as adversarial fine-tuning data to improve ego robustness, measured by collision rate (CR) and overall score (OS) on held-out test scenes; entries report the mean performance over these test scenes. The row labeled "PP" corresponds to the surrogate ego before fine-tuning. The rightmost column shows the average across all base scenarios. For all metrics, ↑ / ↓ indicate that larger/smaller values are preferred, respectively.

| Metric | Algo. | Base Traffic Scenarios | | | | | | | | Avg. |
| --- | --- | --- | --- | --- | --- | --- | --- | --- | --- | --- |
| | | Straight Obstacle | Turning Obstacle | Lane Changing | Vehicle Passing | Red-light Running | Unprotected Left-turn | Right turn | Crossing Negotiation | |
| CR ↓ | PP | 0.950 | 0.850 | 0.775 | 0.940 | 0.750 | 0.850 | 0.850 | 0.860 | 0.854 |
| | ChatScene | **0.050** | **0.025** | 0.550 | **0.000** | 0.050 | 0.400 | **0.000** | **0.000** | 0.134 |
| | ScenePilot | 0.200 | 0.175 | **0.125** | 0.000 | **0.000** | **0.075** | 0.000 | 0.000 | **0.072** |
| OS ↑ | PP | 0.597 | 0.560 | 0.558 | 0.476 | 0.585 | 0.537 | 0.454 | 0.427 | 0.525 |
| | ChatScene | **0.891** | **0.894** | 0.683 | 0.938 | 0.926 | 0.757 | 0.918 | **0.948** | 0.869 |
| | ScenePilot | 0.832 | 0.810 | **0.893** | **0.941** | **0.943** | **0.870** | **0.947** | 0.947 | **0.898** |

phys-invalid frame rate between ScenePilot and ChatScene in pre-collision frames (excluding the final 0.8 s unavoidable collision phase). ChatScene produces a higher fraction of physically invalid frames (i.e., $\sigma_{\text{phys}} < 0$), suggesting that its failures more often involve infeasible interactions before collision. In contrast, ScenePilot keeps the phys-invalid rate much lower while still achieving a higher collision rate, indicating that its failures are less dependent on physically invalid behaviors. We then examine how these frames are distributed in the AV-risk–physical-feasibility space. For each threshold pair $(\phi, \sigma)$, we compute the fraction of frames satisfying both high AV risk and sufficient physical feasibility, and visualize the ScenePilot-to-ChatScene coverage ratio in Fig. 4(b). While ChatScene is mainly concentrated in the lower-left region, ScenePilot covers more high-risk regions on the right side across different feasibility levels. This suggests that ChatScene tends to induce failures under looser feasibility requirements, whereas ScenePilot more broadly explores hazardous yet physically feasible interactions across different safety margins.

### 4.6. Ablation Study

*Table 5.* **Ablation study of ScenePilot.** We report collision rate (CR, ↑), overall score (OS, ↓), and **Gap Coverage Score** (GCS, ↑) on two representative base traffic scenarios. GCS measures the 2D occupancy coverage of the *AV-infeasible yet physically-feasible* gap band in the $(\phi, \sigma)$ space (Appendix C).

| Scenario | Boundary | CR ↑ | OS ↓ | GCS ↑ |
| --- | --- | --- | --- | --- |
| Red-light Running | $\Phi_{\text{risk}}$ only | 0.37 | 0.775 | 0.096 |
| | $\sigma_{\text{phys}}$ only | 0.75 | 0.579 | 0.084 |
| | ScenePilot | **0.93** | **0.488** | **0.209** |
| Right Turn | $\Phi_{\text{risk}}$ only | 0.12 | 0.814 | 0.183 |
| | $\sigma_{\text{phys}}$ only | 0.74 | 0.483 | 0.133 |
| | ScenePilot | **0.91** | **0.399** | **0.351** |

We ablate ScenePilot by enabling each boundary component alone while keeping the same TTC-driven maximization ob-

jective. We consider: (i) $\Phi_{\text{risk}}$ only (constraint $\phi_{\text{risk}} < 1$) and (ii) $\sigma_{\text{phys}}$ only (constraint $\sigma_{\text{phys}} \geq 0$). Table 5 reports CR and OS averaged over SAC/PPO/TD3, together with the Gap Coverage Score (GCS), which measures occupancy coverage of the *AV-infeasible yet physically-feasible* band in the $(\phi, \sigma)$ space (Appendix C). Using $\Phi_{\text{risk}}$ alone yields the weakest safety-criticality and smaller GCS, as risk guidance without feasibility control often remains within AV-feasible interactions and covers a limited subset of failure modes. Using $\sigma_{\text{phys}}$ alone pushes rollouts toward extreme near-boundary behaviors, which appear adversarial but are less consistent across ego controllers and tend to collapse into a narrower region, reducing GCS. Combining both boundaries, ScenePilot attains higher CR, lower OS, and the largest GCS, indicating broader coverage of physically feasible yet AV-challenging interactions without drifting toward overly mild or physically implausible regimes.

## 5. Conclusion

We presented ScenePilot, a controllable boundary-driven framework for generating safety-critical autonomous-driving scenarios. By coupling a physics-based feasibility score $\sigma$ with an online-learned AV-risk predictor $\Phi$ under an $\varepsilon$-constrained multi-objective policy, ScenePilot focuses scenario search on a near-boundary band that is physically solvable yet highly challenging. This design yields more critical and transferable adversarial cases across heterogeneous ego controllers, and can further improve ego robustness through adversarial fine-tuning.

**Limitations.** First, our experiments follow fixed SafeBench base routes, so most diversity concentrates on a short interaction-critical window rather than global trajectory differences. Second, surrogate-driven top-$k$ selection may favor failure modes salient to the surrogate controller; although we replay retained episodes across multiple trained ego controllers, future work should consider more controller-agnostic selection or full-rollout distribution metrics.

## Acknowledgements

This work was supported by the Science and Technology Development Fund of Macau [0123/2022/AFJ, 0081/2022/A2, 0007/2025/RIC, 0122/2024/RIB2, 0215/2024/AGJ, 0074/2025/AMJ, 001/2024/SKL, 0002/2025/EQP], the Research Services and Knowledge Transfer Office, University of Macau [SRG2023-00037-IOTSC, MYRG-GRG2024-00284-IOTSC], the Shenzhen-Hong Kong-Macau Science and Technology Program Category C [SGDX20230821095159012], the Science and Technology Planning Project of Guangdong [2025A0505010016], National Natural Science Foundation of China [52572354], the State Key Lab of Intelligent Transportation System [2024-B001], and the Jiangsu Provincial Science and Technology Program [BZ2024055].

## Impact Statement

This paper presents work aimed at enhancing the safety and reliability of autonomous-driving systems by systematically generating boundary-guided adversarial interactions that rigorously evaluate driving policies. By exposing hard-to-handle corner cases and providing broader, more targeted coverage of challenging behaviors, our approach can support more rigorous safety evaluation and more robust policy improvement. Ultimately, this may contribute to reducing failures in complex traffic situations and improving road safety for autonomous vehicles.

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

# A. Additional Details of ScenePilot

## A.1. Physical Safety Signal $\sigma$

**Physics-limit braking distances.** RSS prescribes appropriate responses for AVs via five common-sense rules:

- **Rule 1:** Maintain a safe longitudinal distance (same-direction and oncoming).

- **Rule 2:** Maintain a safe lateral distance (e.g., merging/cut-in).

- **Rule 3:** Do not fight for the right-of-way.

- **Rule 4:** Be cautious under limited visibility.

- **Rule 5:** If a collision can be avoided without causing another, it must be avoided.

Under Rules 1-2, RSS specifies explicit safe-distance formulas in both longitudinal and lateral directions by Eq. (11). Rule 3 only for posterior responsibility in our fully observable setting. Let $\rho$ denote the reaction time. The two interacting vehicles are indexed by $x \in \{r, f\}$, where $r$ is the rear (left in lateral) vehicle and $f$ is the front (right in lateral) one. The scalars $v_x^{\text{lon}}$ and $v_x^{\text{lat}}$ denote the longitudinal and lateral speed components of vehicle $x$. The maximal accelerations during the reaction window by $a_{x,\max}^{\text{lon,accel}}$ and $a_{x,\max}^{\text{lat,accel}}$, and the conservative RSS braking bounds by $a_{x,\min}^{\text{lon,brake}}$ and $a_{x,\min}^{\text{lat,brake}}$. Define the reaction-updated speeds as $v_{x,\rho}^{\text{lon}} = v_x^{\text{lon}} + \rho a_{x,\max}^{\text{lon,accel}}$ and $v_{x,\rho}^{\text{lat}} = v_x^{\text{lat}} + \rho a_{x,\max}^{\text{lat,accel}}$.

$$
d_{\text{safe}}^{\text{lon,opp}} = \frac{v_r^{\text{lon}} + v_{r,\rho}^{\text{lon}}}{2}\rho + \frac{(v_{r,\rho}^{\text{lon}})^2}{2a_{r,\min}^{\text{lon,brake}}} + \frac{|v_f^{\text{lon}}| + |v_{f,\rho}^{\text{lon}}|}{2}\rho + \frac{(v_{f,\rho}^{\text{lon}})^2}{2a_{f,\min}^{\text{lon,brake}}},
$$

$$
d_{\text{safe}}^{\text{lon,same}} = \left[ v_r^{\text{lon}}\rho + \frac{1}{2}a_{r,\max}^{\text{lon,accel}}\rho^2 + \frac{(v_{r,\rho}^{\text{lon}})^2}{2a_{r,\min}^{\text{lon,brake}}} - \frac{(v_f^{\text{lon}})^2}{2a_{f,\max}^{\text{lon,brake}}} \right]_+,
\tag{11}
$$

$$
d_{\text{safe}}^{\text{lat}} = \mu + \left[ \frac{v_r^{\text{lat}} + v_{r,\rho}^{\text{lat}}}{2}\rho + \frac{(v_{r,\rho}^{\text{lat}})^2}{2a_{r,\min}^{\text{lat,brake}}} - \left( \frac{|v_f^{\text{lat}}| + |v_{f,\rho}^{\text{lat}}|}{2}\rho - \frac{(v_{f,\rho}^{\text{lat}})^2}{2a_{f,\min}^{\text{lat,brake}}} \right) \right]_+.
$$

In RSS's formulation (Eq. (11)), the safe-distance terms are conservative mainly due to: (i) a finite reaction window $\rho$ during which the rear/merging vehicle may continue moving (and potentially accelerating), and (ii) conservative braking assumptions captured by the minimum deceleration bounds $a_{x,\min}^{\text{lon,brake}}$ and $a_{x,\min}^{\text{lat,brake}}$ (together with the lateral buffer $\mu$ for cut-in/merge). Intuitively, this formulation enforces a sufficient separation as long as the follower has the intention to brake within the assumed response model, which can overestimate the distance required to avoid collision in near-miss interactions. For analysis, we therefore also consider an idealized limiting case that removes the reaction delay and characterizes the kinematic boundary under maximal feasible braking, yielding the physics-limit longitudinal/lateral distances in Eq. (2).

The opposite-direction cases follow directly from summing the two axis-wise stopping distances under maximal braking. For the same-direction longitudinal case, however, the tight braking-only boundary depends on the interaction parameters and is therefore written as a maximum of two necessary lower bounds. Consider 1D motion along the longitudinal axis for a rear vehicle $r$ and a front vehicle $f$, with initial speeds $v_r^{\text{lon}}$ and $v_f^{\text{lon}}$, and maximal feasible braking decelerations $a_r := a_{r,\max}^{\text{lon,brake}}$ and $a_f := a_{f,\max}^{\text{lon,brake}}$. Let $d_0$ denote the initial longitudinal gap. One necessary condition is obtained by comparing the two vehicles' stopping distances, which yields

$$
d_0 \geq d_{\text{stop}}^{\text{lon}} = \frac{(v_r^{\text{lon}})^2}{2a_r} - \frac{(v_f^{\text{lon}})^2}{2a_f}.
\tag{12}
$$

A second necessary condition comes from requiring that the inter-vehicle gap never becomes negative during simultaneous braking. With constant decelerations, the relative distance evolves as

$$
d(t) = d_0 + (v_f^{\text{lon}} - v_r^{\text{lon}})t + \frac{1}{2}(a_r - a_f)t^2.
\tag{13}
$$

When a closing interaction occurs ($v_r^{\text{lon}} > v_f^{\text{lon}}$) and the rear can decelerate more strongly ($a_r > a_f$), the quadratic in (13) attains its minimum at

$$t^\star = \frac{v_r^{\text{lon}} - v_f^{\text{lon}}}{a_r - a_f}, \tag{14}$$

which is exactly the time when the two longitudinal velocities become equal. Requiring the minimum gap to remain nonnegative, $d(t^\star) \geq 0$, gives

$$d_0 \geq d_{\text{equal}}^{\text{lon}} = \frac{(v_r^{\text{lon}} - v_f^{\text{lon}})^2}{2(a_r - a_f)}. \tag{15}$$

To make the parameter dependence explicit, introduce the dimensionless ratios $\kappa := \frac{a_r}{a_f}$, $\eta := \frac{v_r^{\text{lon}}}{v_f^{\text{lon}}}$. In the common closing regime $v_r^{\text{lon}} > v_f^{\text{lon}}$ and $a_r > a_f$ (i.e., $\eta > 1$ and $\kappa > 1$), the two bounds in (12)–(15) can be compared as

$$\begin{aligned}
d_{\text{stop}}^{\text{lon}} - d_{\text{equal}}^{\text{lon}} &= \frac{(v_f^{\text{lon}})^2}{2a_f} \left( \frac{\eta^2}{\kappa} - 1 - \frac{(\eta - 1)^2}{\kappa - 1} \right) \\
&= \frac{(v_f^{\text{lon}})^2}{2a_f} \cdot \frac{(\eta^2 - \kappa)(\kappa - 1) - \kappa(\eta - 1)^2}{\kappa(\kappa - 1)}.
\end{aligned} \tag{16}$$

Thus, depending on $(v_r^{\text{lon}}, v_f^{\text{lon}}, a_r, a_f)$, either $d_{\text{stop}}^{\text{lon}}$ or $d_{\text{equal}}^{\text{lon}}$ can be the tighter constraint. Since collision avoidance under braking-only responses must satisfy both necessary conditions, and the active one varies with the interaction parameters, we adopt the unified expression

$$d_{\text{same}}^{\text{lon}} = \max\left( d_{\text{stop}}^{\text{lon}}, d_{\text{equal}}^{\text{lon}} \right), \tag{17}$$

which matches the same-direction longitudinal term in Eq. (2). Finally, this axis-wise braking-limit distance provides a tighter notion of what is physically avoidable: if $d_{\text{actual}} < d_{\text{limit}}$ on an axis, then a collision is unavoidable under braking-only responses along that axis.

**Physical safety signal $\sigma$.** As discussed in Sec. 3.2, the physics-limit distance $d_{\text{limit}}$ characterizes the boundary of braking-only recoverability along a given axis. However, near-boundary interactions can still be solvable via orthogonal maneuvers (e.g., evasive steering / lane change) when sufficient time and dynamic margin exist. We therefore aggregate the axis-wise residual margins into a single physical safety signal $\sigma$ in Eq. (3). Below we detail how each term in Eq. (3) is computed.

We first compute the adversary's relative position in the ego-fixed body frame. Let $p_{\text{ego}} = [x_{\text{ego}}, y_{\text{ego}}]^\top$ and $p_{\text{adv}} = [x_{\text{adv}}, y_{\text{adv}}]^\top$ be the world-frame centers, and let $R(-\psi_{\text{ego}})$ rotate world coordinates into the ego frame. Then

$$\begin{bmatrix} d_x^{\text{actual}} \\ d_y^{\text{actual}} \end{bmatrix} = R(-\psi_{\text{ego}}) (p_{\text{adv}} - p_{\text{ego}}), \qquad R(-\psi_{\text{ego}}) = \begin{bmatrix} \cos\psi_{\text{ego}} & \sin\psi_{\text{ego}} \\ -\sin\psi_{\text{ego}} & \cos\psi_{\text{ego}} \end{bmatrix}. \tag{18}$$

To account for finite vehicle sizes and relative yaw, we use the envelope terms in Eq. (3). Given $\Delta\psi = \psi_{\text{adv}} - \psi_{\text{ego}}$, define

$$\begin{aligned}
s_x(\Delta\psi) &= L_{\text{ego}} + |\cos\Delta\psi| L_{\text{adv}} + |\sin\Delta\psi| W_{\text{adv}}, \\
s_y(\Delta\psi) &= W_{\text{ego}} + |\cos\Delta\psi| W_{\text{adv}} + |\sin\Delta\psi| L_{\text{adv}},
\end{aligned} \tag{19}$$

and the signed edge-to-edge clearances

$$\Delta d_x = |d_x^{\text{actual}}| - s_x(\Delta\psi), \qquad \Delta d_y = |d_y^{\text{actual}}| - s_y(\Delta\psi). \tag{20}$$

if the edge-to-edge clearance on an axis $a \in \{x, y\}$ satisfies $\Delta d_a < d_{\text{limit},a}$, We estimate the time before the first axis-wise collision event using the instantaneous closing speeds:

$$t_x = \frac{[\Delta d_x]_+}{[\Delta v_x]_+}, \qquad t_y = \frac{[\Delta d_y]_+}{[\Delta v_y]_+}, \qquad t = \min(t_x, t_y), \tag{21}$$

where $[\cdot]_+$ denotes $\max(0, \cdot)$. This TTC (time to collision) is a local kinematic estimate computed from the current relative state. Formally, letting $d_c(\tau)$ be the edge-to-edge clearance on axis $c \in \{x, y\}$ and $\Delta v_c(\tau)$ be the corresponding closing speed, the collision time under the instantaneous state satisfies

$$0 \approx d_c(t) = d_c(0) - \int_0^t [\Delta v_c(\tau)]_+ \, d\tau. \tag{22}$$

Because TTC is evaluated as an instantaneous per-frame estimate, we approximate the closing speed as locally constant over the short prediction horizon around the current frame, i.e., $[\Delta v_c(\tau)]_+ \approx [\Delta v_c(0)]_+$. This gives

$$t \approx \frac{[d_c(0)]_+}{[\Delta v_c(0)]_+}, \tag{23}$$

which yields Eq. (21) after substituting the axis-specific clearances $d_x(0) = |d_x^{\text{actual}}| - s_x(\Delta\psi)$ and $d_y(0) = |d_y^{\text{actual}}| - s_y(\Delta\psi)$ and taking the earliest axis-wise event via $t = \min(t_x, t_y)$. The estimate is refreshed at every simulator frame when computing $\sigma$, so the local approximation is repeatedly refreshed along the rollout and acts as a discrete-time accumulation of the evolving relative motion. In contrast, computing TTC by assuming future maximal braking would correspond to replacing $[\Delta v_c(\tau)]_+$ in (22) with a braking-shaped profile, which can systematically enlarge $t$ and thus over-credit recoverability near the boundary.

Let $c = \arg\min\{t_x, t_y\}$ be the first-colliding axis and $n = \{x, y\} \setminus \{c\}$ its orthogonal axis. We set $l_c = 0$ and compensate only along the orthogonal axis by the reachable displacement within time $t$:

$$l_n = \tfrac{1}{2} a_n^{\text{rel}} t^2 \mathbf{1}\{\Delta v_n > 0\}, \tag{24}$$

so that $(l_x, l_y) = (0, l_n)$ if $c = x$ and $(l_x, l_y) = (l_n, 0)$ if $c = y$. This $\frac{1}{2}at^2$ form follows from the constant-acceleration kinematic relation for displacement, and we use it as an upper bound under a physics-limit assumption: once the imminent-collision axis is identified, the most favorable solvable behavior is to generate separation along the orthogonal axis with maximal feasible relative acceleration. Accordingly, $l_n$ can be interpreted as the additional clearance that could be created before the earliest collision time, while $\mathbf{1}\{\Delta v_n > 0\}$ avoids crediting compensation when the orthogonal gap is not closing.

Finally, letting $d_x^{\text{limit}}$ and $d_y^{\text{limit}}$ be the axis-wise physics-limit distances from Eq. (2) (projected onto the ego $x/y$ axes), Eq. (3) can be evaluated equivalently in the explicit scalar form

$$\sigma = 1 - \left( \left[ \frac{\left[ d_x^{\text{limit}} - \Delta d_x - l_x \right]_+}{d_x^{\text{limit}}} \right]^p + \left[ \frac{\left[ d_y^{\text{limit}} - \Delta d_y - l_y \right]_+}{d_y^{\text{limit}}} \right]^p \right)^{1/p}. \tag{25}$$

In our implementation, we set $p = 2$, which corresponds to an $\ell_2$ aggregation of the normalized axis-wise gaps and yields an elliptical level-set geometry in the $(x, y)$ plane (a convenient surrogate for the vehicle footprint when combining longitudinal and lateral margins). By construction, $\sigma > 0$ indicates physically feasible frames, $\sigma = 0$ lies on the kinematic boundary, and $\sigma < 0$ indicates physically infeasible frames where no admissible combination of braking and steering can prevent a collision.

## A.2. AV Risk Signal $\phi$

For per-frame AV risk estimation, a lightweight predictor is needed that can be queried at every frame with negligible overhead. Instead of introducing an additional planner or a heavy safety module, aa Bellman-style formulation is trained to approximate the discounted collision-to-go return via Eq. (26), which is theoretically sufficient.

$$\tilde{\Phi}(s_t) = \mathbb{E}\left[ \mathbb{K}\{\mathcal{C}_t\} + \gamma \, \tilde{\Phi}(s_{t+1}) \,\middle|\, s_t \right]. \tag{26}$$

However, collisions are rare and most frames satisfy $\mathbb{K}\{\mathcal{C}_t\} = 0$. In long non-collision segments, the recursion in Eq. (26) is dominated by bootstrap term $\gamma \, \tilde{\Phi}(s_{t+1})$, yielding weak learning signal for earlier frames and causing the estimated risk to drift. We therefore use the shaped target in Eq. (5) by inserting a potential-based shaping term $\gamma F(s_{t+1}) - F(s_t)$ into the per-step part of the TD target. Concretely, along a rollout $s_0 \to s_1 \to s_2 \to \cdots$, the shaping contributions at successive steps take the form

$$\gamma F(s_1) - F(s_0),$$
$$\gamma F(s_2) - F(s_1),$$
$$\gamma F(s_3) - F(s_2), \ \ldots$$

so each step is given an additional dense signal that depends on how the state potential changes. That is, the per-step signal is replaced from $\mathbb{K}\{\mathcal{C}_t\}$ to the shaped form $\mathbb{K}\{\mathcal{C}_t\} + \gamma F(s_{t+1}) - F(s_t)$. Accordingly, the shaped per-step reward as

$$\hat{r}_t = \mathbb{K}\{\mathcal{C}_t\} + \gamma F(s_{t+1}) - F(s_t). \tag{27}$$

so the TD target in Eq. (5) can be written as $y_t = \hat{r}_t + \gamma\hat{\Phi}(s_{t+1})$. The corresponding discounted shaped return is

$$\hat{G} = \sum_{t=0}^{T}\gamma^t\hat{r}_t = \sum_{t=0}^{T}\gamma^t\mathbb{K}\{\mathcal{C}_t\} + \sum_{t=0}^{T}\gamma^t\big(\gamma F(s_{t+1}) - F(s_t)\big). \tag{28}$$

The shaping sum telescopes

$$\sum_{t=0}^{T}\gamma^t\big(\gamma F(s_{t+1}) - F(s_t)\big) = \sum_{t=0}^{T}\gamma^{t+1}F(s_{t+1}) - \sum_{t=0}^{T}\gamma^t F(s_t) \tag{29}$$

$$= \gamma^{T+1}F(s_{T+1}) - F(s_0).$$

If $F$ is bounded and $\gamma \in (0,1)$, then $\lim_{T\to\infty}\gamma^{T+1}F(s_{T+1}) = 0$, and thus

$$\hat{G} = \underbrace{\sum_{t=0}^{\infty}\gamma^t\mathbb{K}\{\mathcal{C}_t\}}_{G} - F(s_0). \tag{30}$$

Therefore, potential shaping preserves the underlying collision-to-go objective and only introduces a state-dependent constant shift in the return at the starting state. Taking conditional expectation yields the shifted value-function relation

$$\hat{\Phi}(s) = \mathbb{E}[\hat{G} \mid s_0 = s] = \underbrace{\mathbb{E}[G \mid s_0 = s]}_{\Phi(s)} - F(s), \tag{31}$$

which implies $\Phi(s) = \hat{\Phi}(s) + F(s)$. Consequently, when the critic is trained with Eq. (5), we recover the AV-risk signal on the original scale by adding back the potential

$$\phi_t := \Phi(s_t) = \hat{\Phi}(s_t) + F(s_t). \tag{32}$$

### A.3. Feasibility Threshold $\varepsilon$

ScenePilot uses the feasibility threshold $\varepsilon$ to control the safety margin from the physical feasibility boundary. Since $\sigma = 0$ corresponds to the physical feasibility boundary, small positive $\varepsilon$ values probe near-boundary interactions where failures can still be induced, while larger $\varepsilon$ values impose stricter feasibility margins and move the search farther inside the feasible region. As this margin increases, the generated interactions move farther away from the physical limit and are more likely to fall within the AV policy capability region, making collision-inducing cases progressively sparser. Thus, $\varepsilon$ is not merely a training coefficient, but the control variable that determines how ScenePilot sweeps the feasible-yet-adversarial boundary band.

We empirically observe that the density of collision-inducing cases is highly non-uniform across different feasibility-threshold ranges. Table 6 reports the collision rate observed under different $\varepsilon$ intervals during training for some generated scenarios. The trend shows that, as $\varepsilon$ increases, collision-inducing cases become progressively rarer. This observation motivates both the use of non-uniform threshold spacing and the choice of $\varepsilon_{\max} = 0.35$, since larger feasibility margins yield almost no informative failure cases.

*Table 6.* Collision rate under different feasibility-threshold intervals during training. Larger $\varepsilon$ values impose stricter feasibility margins, making collision-inducing cases progressively sparser.

| $\varepsilon$ **interval** | **Collision Rate** |
|---|---|
| $0 \sim 0.1$ | 0.376 |
| $0.1 \sim 0.2$ | 0.258 |
| $0.2 \sim 0.3$ | 0.107 |
| $0.3 \sim 0.35$ | 0.070 |
| $0.35$ | 0.000 |

Based on this non-uniform boundary structure, we adopt a Gaussian-shaped schedule rather than uniformly spaced thresholds. The schedule allocates denser constraint levels near the physical feasibility boundary and coarser levels farther inside the feasible region. This design enables more fine-grained probing of the near-boundary region, where small changes in $\varepsilon$ can substantially affect whether a rollout remains physically feasible and still collision-inducing.

Concretely, we predefine $N$ constraint levels by sampling evenly spaced points on a truncated standard-normal axis and mapping them to $[0, \varepsilon_{\max}]$ through the standard-normal cumulative distribution function:

$$\varepsilon_i = \varepsilon_{\max} \cdot \frac{F_{\mathcal{N}}(u_i) - F_{\mathcal{N}}(u_{\min})}{F_{\mathcal{N}}(u_{\max}) - F_{\mathcal{N}}(u_{\min})}, \quad u_i = u_{\min} + \frac{i-1}{N-1}(u_{\max} - u_{\min}), \quad i = 1, \ldots, N, \tag{33}$$

where $F_{\mathcal{N}}(\cdot)$ denotes the cumulative distribution function of the standard normal distribution. This produces a sequence in which small $\varepsilon$ values are explored with finer granularity, while larger $\varepsilon$ values are explored more sparsely.

During training, we switch the active threshold every 100 episodes, cycling through $\{\varepsilon_i\}_{i=1}^{N}$ until reaching $\varepsilon_{\max} = 0.35$. Each time a threshold level is revisited, optimization resumes from the checkpoint previously saved for that same threshold $\varepsilon_i$. After reaching $\varepsilon_{\max}$, we restart the sweep from small $\varepsilon$ and continue this threshold-specific optimization process. This cyclic sweep stabilizes training while repeatedly revisiting the near-boundary feasible region under different feasibility-margin requirements.

## B. Experiment Details

We adopt the publicly released AV controllers provided by ChatScene (Zhang et al., 2024) as our surrogate AVs. Concretely, we use SAC as the default AV during training, and evaluate on a diverse set of AV planners including SAC and PPO (with two SAC checkpoints of different training configurations) as well as TD-based baselines. All AV checkpoints are directly taken from the official ChatScene releasing and are used as-is without any additional fine-tuning or modification.

### B.1. Physical Feasibility Parameters

We compute the physical-feasibility signal $\sigma$ online at every timestep in CARLA. Vehicle geometry is obtained from the CARLA actor bounding box at runtime, and all checks follow the simulator step size $\Delta t$. Table 7 summarizes the constants used in Eq. 3. We aggregate axis-wise residual margins into $\sigma$ using a $p$-norm with $p = 2$.

*Table 7.* Physical-feasibility parameters used for computing $\sigma$.

| Parameter | Value |
|---|---|
| Simulator step $\Delta t$ | $0.1\,\mathrm{s}$ |
| Vehicle geometry | CARLA actor bounding box (runtime) |
| $p$-norm (aggregation) $p$ | 2 |
| TTC lower bound | $1 \times 10^{-4}$ |
| Far-distance cap | 10000 |
| Minimum lateral safe distance | 0.30 |
| Same-direction yaw threshold | $30°$ |
| AV max longitudinal decel | $3.0\,\mathrm{m/s^2}$ |
| NPC max longitudinal decel | $4.0\,\mathrm{m/s^2}$ |
| AV max lateral decel | $2.0\,\mathrm{m/s^2}$ |
| NPC max lateral decel | $1.5\,\mathrm{m/s^2}$ |

### B.2. AV Risk Network Training

**Data Source.** To learn a stable per-frame AV risk predictor, it is important to expose the critic to sufficient safety-critical interactions, since collisions are sparse under normal traffic. Within the SafeBench benchmark, we train a PPO-based scenario policy to actively generate hazardous encounters, and simultaneously update the AV risk network online using the same interaction stream. Specifically, the PPO scenario policy is optimized with a composite reward consisting of (i) our physical-safety signal (Sec. A.1) and (ii) a TTC-based adversarial reward, which together encourage near-miss and boundary

interactions while maintaining meaningful traffic dynamics. In practice, each scenario is run for 50k environment steps; we train across 8 scenarios, resulting in 400k interaction steps that jointly drive scenario learning and AV-risk learning.

**Training details.** We train a lightweight risk critic that maps a compact 6-D interaction feature to a scalar risk value in $[0, 1]$. To stabilize learning under sparse collisions, we introduce a distance-based potential $F(s) = k \cdot \mathrm{inv\_d}(s)$. The critic is optimized with TD(0) bootstrapping using a slowly-updated target network: $y_t = \mathrm{clip}\Big(r_t + (1 - \mathrm{done}_t)\gamma \hat{\Phi}_{\mathrm{tgt}}(s_{t+1}), 0, 1\Big)$, where $r_t$ is a dense signal composed of a collision indicator and a potential-based shaping term. We minimize a weighted binary cross-entropy with clipped soft targets in $[0, 1]$ to address the severe imbalance between risky and non-risky frames, up-weighting high-risk targets ($y_t > 0.85$) by $w_+$. All hyperparameters and architectural choices are reported in Table 8.

*Table 8.* Hyperparameters for AV risk network training.

| Parameter | Value |
|---|---|
| Training benchmark | SafeBench (online during PPO scenario rollouts) |
| Training budget | 8 scenarios $\times$ 50k steps (400k env steps) |
| Input feature dim. | 6 |
| Approximation function | MLP |
| Hidden units per layer | 128 |
| Hidden activation | ReLU |
| Output activation | Sigmoid |
| Optimizer | Adam |
| Learning rate | $1 \times 10^{-3}$ |
| Discount factor $\gamma$ | 0.95 |
| Loss | Weighted BCE |
| High-risk weight $w_+$ | 50 (for target $> 0.85$) |
| Potential scale $k$ (`k_inv_d`) | 1 |
| Target network | Polyak (soft) update |
| Polyak coefficient $\tau$ | 0.01 |

### B.3. ScenePilot Policy Training

**Training details.** We train the ScenePilot scenario policy on the SafeBench benchmark (CARLA) using an on-policy policy-gradient optimizer, while keeping the surrogate AV controller fixed to the publicly released checkpoints from ChatScene (without any additional fine-tuning). The trainable scenario policy is a small MLP-based actor-critic with only about 55K trainable parameters, so ScenePilot does not rely on large-scale neural optimization. All experiments were run on a server with 2$\times$ RTX 4090 GPUs; under this setup, we typically train the 10 routes of one base scenario in parallel, and the wall-clock training time for a single base scenario is about 15 hours. Unless otherwise stated, we set the discount factor $\gamma = 0.99$, GAE $\lambda = 0.95$, PPO clipping parameter $\epsilon_{\mathrm{clip}} = 0.25$, Adam learning rate $1 \times 10^{-4}$, batch size 2048, one update epoch per iteration, and entropy coefficient 0.03. The feasibility threshold follows the Gaussian-shaped schedule in Appendix A.3, with $N = 8$ predefined constraint levels. Figures 5 and 6 report the aggregated training curves over all route-wise runs, illustrating the typical periodic dynamics induced by the cyclic constraint sweep. All hyperparameters are summarized in Table 9.

### B.4. Scenario Sampling

Each base scenario contains 10 predefined routes in SafeBench. Using the fixed SAC ego policy released by ChatScene (Zhang et al., 2024) as the surrogate controller, we train ScenePilot independently on each route for 3000 episodes under the above cyclic $\varepsilon$ sweep to obtain route-specialized adversarial behaviors. For each (base scenario, route) pair, we keep the top-10 most safety-critical rollouts, resulting in 100 episodes per base scenario (10 routes). To avoid dependence on the surrogate, we report final results by replaying these episodes under three different ego controllers (PPO, SAC, and TD3) released by ChatScene and averaging over controllers and routes.

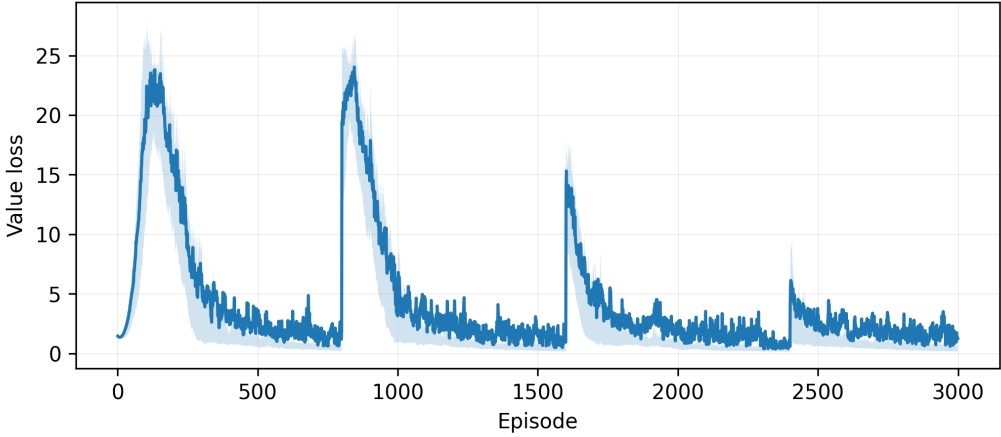

*Figure 5.* Aggregated value loss during ScenePilot training.

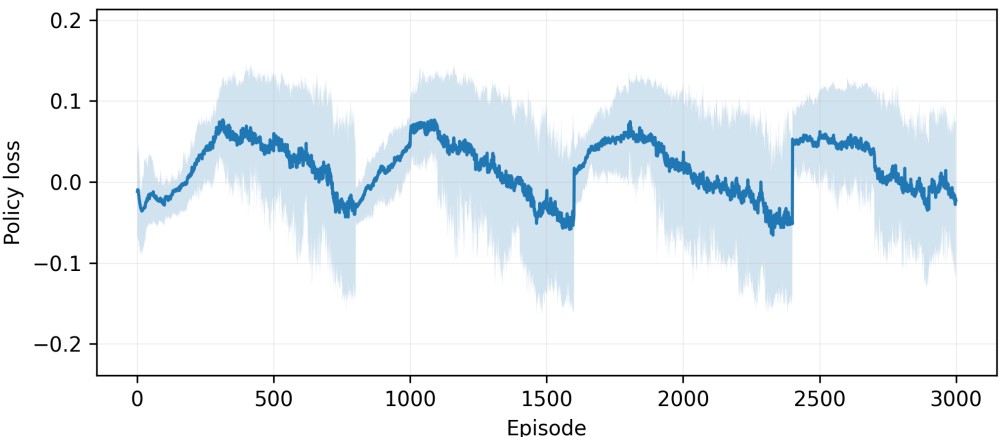

*Figure 6.* Aggregated policy loss during ScenePilot training.

### B.5. AV Policy Fine-tuning

We follow the adversarial fine-tuning procedure of ChatScene to adapt the surrogate AV policy under generated adversarial scenarios. Concretely, we start from the publicly released SAC-trained surrogate AV checkpoint from Chatscene and fine-tune it in the same simulator setting without modifying the architecture. We run fine-tuning for 500 epochs, where one epoch corresponds to one full simulation rollout in one scene. Both the policy learning rate and the Q-value learning rate are set to $1 \times 10^{-4}$. To mitigate the extreme imbalance between collision and non-collision experiences during adversarial interaction, we maintain two replay buffers: a non-collision buffer of size $20,000$ and a collision buffer of size $200$. We use a batch size of 512; for each gradient step, 80% of samples are drawn from the non-collision buffer and 20% from the collision buffer. If the collision buffer contains fewer than $0.2 \times 512$ samples, we include all available collision samples and fill the remainder with non-collision samples. The entropy regularization coefficient is fixed to $0.01$, the discount factor is set to $\gamma = 0.99$, and the Q-ensemble critic uses two Q-networks, consistent with ChatScene.

To address the failure mode that overly long fine-tuning may induce a degenerate stopping strategy, we adopt the same checkpoint-selection protocol as ChatScene. Specifically, to obtain a faithful estimate of adversarial robustness, we evaluate all fine-tuned checkpoints after 100 epochs with a stride of 50 epochs, and report the best checkpoint. The selected checkpoint is defined as the one achieving the lowest collision rate while maintaining a reasonable route completion rate ($> 0.3$), since adversarial events in our benchmark typically occur after the first 30% of route progress.

*Table 9.* Hyperparameters for ScenePilot scenario-policy training.

| Parameter | Value |
|---|---|
| Training benchmark | SafeBench (CARLA, on-policy rollouts) |
| Surrogate AV | SAC |
| Scenario policy | ScenePilot |
| Discount factor $\gamma$ | 0.99 |
| GAE $\lambda$ | 0.95 |
| Clip $\epsilon_{\text{clip}}$ | 0.25 |
| Batch size | 2048 |
| Update epochs per iteration | 1 |
| Entropy coefficient | 0.03 |
| Optimizer / LR | Adam / $1 \times 10^{-4}$ |
| Gradient clipping | 0.5 |
| Constraint | $\max J_{\text{risk}}$ s.t. $J_\sigma \geq \varepsilon$ |
| $\varepsilon$ schedule | Gaussian-spaced levels (Eq. 33) |
| $N$ | 8 |
| $\varepsilon_{\max}$ | 0.35 |
| Switch frequency | every 100 episodes |
| After reaching $\varepsilon_{\max}$ | repeat schedule; continue from current weights |
| Checkpoint organization | indexed by active $\varepsilon$ |

## C. Result Details

**Detailed metrics.** We adopt the evaluation metrics from Chatscene (Zhang et al., 2024) and briefly summarize the intuition of each metric: CR (collision rate) evaluates the rate of collisions, reflecting accident-avoidance capability; RR (red-light running frequency) measures how often the ego vehicle runs red lights; SS (stop-sign violation frequency) measures failures to stop at stop signs; OR (out-of-road distance) quantifies the average distance driven out of road, indicating lane/road discipline; RF (route-following stability) examines how stably the ego vehicle follows its planned route; Comp (route completion) reports the percentage of the route completed; TS (time spent) evaluates the time efficiency to finish a route; ACC (average acceleration) measures driving smoothness; YV (average yaw velocity) reflects turning/handling intensity; and LI (lane invasion frequency) measures lane-keeping accuracy. The overall score (OS) is an aggregated metric combining the above factors, and we use the same weighting scheme as ChatScene.

In addition, we report **Gap Coverage Score (GCS)**, which quantifies how broadly the generated interactions populate the *AV-infeasible yet physically-feasible* gap band in the $(\phi, \sigma)$ space. Given frame-level pairs $\mathcal{S} = \{(\phi_t, \sigma_t)\}_{t=1}^T$ collected along rollouts of the generated critical window, we define the *gap frames* as those satisfying physical feasibility while coming from failure episodes:

$$\mathcal{S}_{\text{gap}} = \{(\phi_t, \sigma_t) \in \mathcal{S} \mid \sigma_t > 0 \ \wedge \ \texttt{collision\_rate}(\text{episode}(t)) > 0\}. \tag{34}$$

We discretize $[0, 1] \times [0, 1]$ in the $(\phi, \sigma)$ plane into a fixed $K \times K$ grid (with $K = 40$ by default), and denote the set of grid cells as $\mathcal{C}_K$. A cell is marked as occupied if it contains at least one point from $\mathcal{S}_{\text{gap}}$, i.e.,

$$\mathcal{O}(\mathcal{S}_{\text{gap}}) = \{c \in \mathcal{C}_K \mid \exists (\phi, \sigma) \in \mathcal{S}_{\text{gap}} \text{ s.t. } (\phi, \sigma) \in c\}. \tag{35}$$

Then the Gap Coverage Score is defined as

$$\text{GCS} = \frac{|\mathcal{O}(\mathcal{S}_{\text{gap}})|}{|\mathcal{C}_K|} = \frac{|\mathcal{O}(\mathcal{S}_{\text{gap}})|}{K^2}. \tag{36}$$

Larger GCS indicates that the generator covers a wider portion of the feasible gap, i.e., it discovers more diverse physically feasible yet AV-challenging interaction regimes, rather than collapsing to overly mild or overly extreme behaviors.

**Results.** We provide a detailed assessment of adversarial performance under multiple surrogate AV controllers. Following the main protocol, we evaluate each scene set using three ego-vehicle policies trained with different RL algorithms, namely SAC, PPO, and TD3, to test whether the generated adversarial scenes generalize across diverse AV training recipes. Concretely, we report the *collision rate* (CR) and the *overall score* (OS) for each test ego vehicle under the same base traffic scenarios. Detailed statistics of CR are summarized in Table 10, while the OS results are reported in Table 11. The evaluation protocol details (including metrics, scoring rules) follow the official evaluation settings of ChatScene.

Table 12 provides a multi-level diagnostic by decomposing the SafeBench overall score (OS) into safety-, functionality-, and etiquette-related indicators. ScenePilot achieves the highest collision rate (CR) and the best OS, while keeping several non-collision proxies comparatively low. This is expected because our scenario generation operates on fixed SafeBench base routes and does not explicitly optimize traffic-rule indicators such as red-light or stop-sign violations (RR/SS); instead, it targets safety-criticality through adversarial interactions between the ego and non-ego agents. Even under this constraint, ScenePilot still establishes a new overall best result under the standard SafeBench evaluation, indicating that physics-guided boundary shaping combined with AV-risk-aware optimization yields more transferable and genuinely safety-critical scenarios. Consequently, these results suggest that ScenePilot's increased safety-criticality primarily arises from interaction-driven, near-boundary failure cases—i.e., scenarios that expose control-limit and collision-prone interactions—rather than from trivial rule-breaking or unstable driving behaviors that can artificially inflate risk.

Overall, the results indicate that our method yields more transferable adversarial scenes and achieves consistently stronger average performance across different ego-vehicle training algorithms. We further provide visualizations of the generated adversarial interactions. These examples illustrate representative near-boundary behaviors and failure cases across different base scenarios.

*Table 10.* **Collision Rate (CR) Performance Across Different Models.** We report the mean CR for each model across base traffic scenarios. Higher values of CR (↑) indicate stronger adversarial effectiveness. Bold denotes the best performance for each scenario.

| Ego | Algo. | Straight Obstacle | Turning Obstacle | Lane Changing | Vehicle Passing | Red-light Running | Unprotected Left-turn | Right turn | Crossing Negotiation | Avg. |
|---|---|---|---|---|---|---|---|---|---|---|
| | | | | | Base Traffic Scenarios | | | | | |
| SAC (4D) | LC | 0.40 | 0.11 | 0.60 | 0.80 | 0.92 | 0.86 | 0.70 | 0.75 | 0.642 |
| | AS | 0.53 | 0.40 | 0.75 | **0.90** | 0.62 | 0.85 | 0.21 | 0.53 | 0.599 |
| | CS | 0.53 | 0.68 | 0.67 | **0.90** | 0.76 | 0.90 | 0.69 | 0.68 | 0.725 |
| | AT | 0.73 | 0.48 | 0.77 | **0.90** | **1.00** | 0.97 | 0.76 | 0.90 | 0.814 |
| | ChatScene | 0.94 | 0.73 | 0.92 | 0.81 | 0.70 | 0.88 | 0.84 | 0.78 | 0.825 |
| | ScenePilot | **0.97** | **0.90** | **1.00** | 0.86 | **1.00** | **0.99** | **0.90** | **0.95** | **0.946** |
| PPO (4D) | LC | 0.09 | 0.11 | **1.00** | **1.00** | 0.22 | 0.20 | 0.28 | 0.00 | 0.363 |
| | AS | 0.39 | 0.19 | **1.00** | **1.00** | 0.67 | 0.61 | 0.62 | **0.92** | 0.675 |
| | CS | 0.22 | 0.61 | **1.00** | **1.00** | 0.47 | 0.37 | 0.52 | 0.44 | 0.579 |
| | AT | 0.10 | 0.11 | 0.98 | 0.87 | 0.13 | 0.21 | 0.03 | 0.05 | 0.310 |
| | ChatScene | **0.87** | 0.61 | 0.97 | 0.98 | 0.89 | 0.52 | 0.74 | 0.91 | 0.811 |
| | ScenePilot | 0.84 | **0.79** | 0.99 | **1.00** | **0.96** | **0.83** | **0.87** | 0.80 | **0.885** |
| TD3 (4D) | LC | 0.42 | 0.06 | **1.00** | 0.70 | **1.00** | **1.00** | 0.79 | **1.00** | 0.746 |
| | AS | 0.60 | 0.39 | 0.83 | 0.70 | 0.41 | 0.65 | 0.03 | 0.26 | 0.484 |
| | CS | 0.61 | 0.53 | **1.00** | 0.70 | 0.67 | 0.80 | 0.83 | 0.67 | 0.726 |
| | AT | 0.67 | 0.35 | 0.59 | 0.70 | **1.00** | 0.85 | **0.99** | 0.90 | 0.756 |
| | ChatScene | 0.87 | 0.75 | 0.96 | **0.99** | 0.77 | 0.86 | 0.75 | 0.90 | **0.856** |
| | ScenePilot | **0.88** | **0.82** | **1.00** | 0.81 | 0.84 | 0.85 | 0.97 | 0.63 | 0.850 |

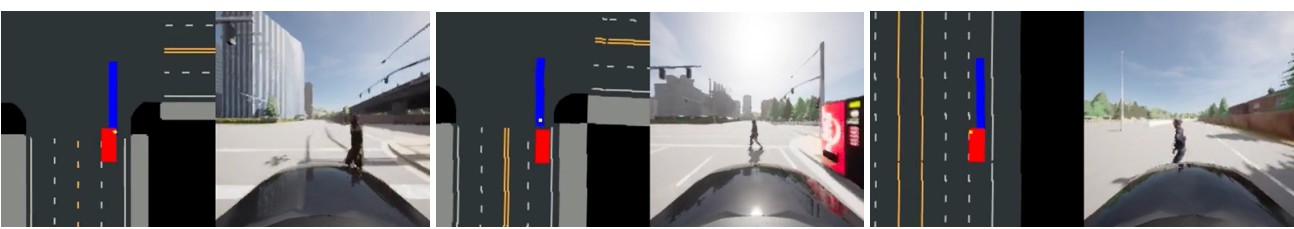

Straight Obstacle

*Table 11.* **Overall Score (OS) Performance Across Different Models.** We report the mean OS for each model across base traffic scenarios. Lower values of OS (↓) are preferable. Bold denotes the best performance for each scenario.

| Ego | Algo. | Base Traffic Scenarios | | | | | | | | Avg. |
| --- | --- | --- | --- | --- | --- | --- | --- | --- | --- | --- |
| | | Straight Obstacle | Turning Obstacle | Lane Changing | Vehicle Passing | Red-light Running | Unprotected Left-turn | Right turn | Crossing Negotiation | |
| SAC (4D) | LC | 0.716 | 0.824 | 0.617 | 0.515 | 0.491 | 0.521 | 0.497 | 0.500 | 0.585 |
| | AS | 0.663 | 0.673 | 0.552 | **0.466** | 0.650 | 0.538 | 0.746 | 0.617 | 0.613 |
| | CS | 0.661 | 0.532 | 0.569 | **0.466** | 0.578 | 0.508 | 0.503 | 0.539 | 0.544 |
| | AT | 0.565 | 0.633 | 0.546 | **0.466** | 0.462 | 0.475 | 0.461 | **0.423** | 0.504 |
| | ChatScene | **0.450** | 0.505 | **0.451** | 0.489 | 0.582 | 0.492 | 0.426 | 0.461 | 0.482 |
| | ScenePilot | 0.471 | **0.471** | 0.454 | 0.479 | **0.460** | **0.457** | 0.406 | **0.396** | **0.456** |
| PPO (4D) | LC | 0.858 | 0.823 | 0.457 | 0.445 | 0.850 | 0.858 | 0.702 | 0.887 | 0.735 |
| | AS | 0.726 | 0.780 | 0.457 | 0.444 | 0.617 | 0.646 | 0.535 | **0.408** | 0.577 |
| | CS | 0.806 | 0.570 | 0.468 | 0.444 | 0.722 | 0.771 | 0.582 | 0.656 | 0.627 |
| | AT | 0.853 | 0.819 | **0.439** | 0.487 | 0.896 | 0.852 | 0.826 | 0.861 | 0.754 |
| | ChatScene | **0.497** | 0.578 | 0.444 | **0.421** | 0.503 | 0.705 | 0.504 | 0.417 | 0.509 |
| | ScenePilot | 0.532 | **0.527** | 0.460 | 0.435 | **0.470** | **0.536** | 0.420 | 0.473 | **0.482** |
| TD3 (4D) | LC | 0.708 | 0.843 | 0.442 | 0.561 | **0.461** | **0.466** | 0.447 | **0.378** | 0.538 |
| | AS | 0.631 | 0.668 | 0.511 | 0.561 | 0.757 | 0.638 | 0.834 | 0.755 | 0.669 |
| | CS | 0.627 | 0.599 | 0.430 | 0.561 | 0.622 | 0.559 | 0.430 | 0.542 | 0.546 |
| | AT | 0.587 | 0.689 | 0.629 | 0.561 | 0.462 | 0.534 | **0.348** | 0.423 | 0.529 |
| | ChatScene | **0.462** | **0.483** | **0.407** | **0.410** | 0.527 | 0.483 | 0.493 | 0.385 | **0.456** |
| | ScenePilot | 0.510 | 0.514 | 0.461 | 0.501 | 0.533 | 0.528 | 0.371 | 0.564 | 0.498 |

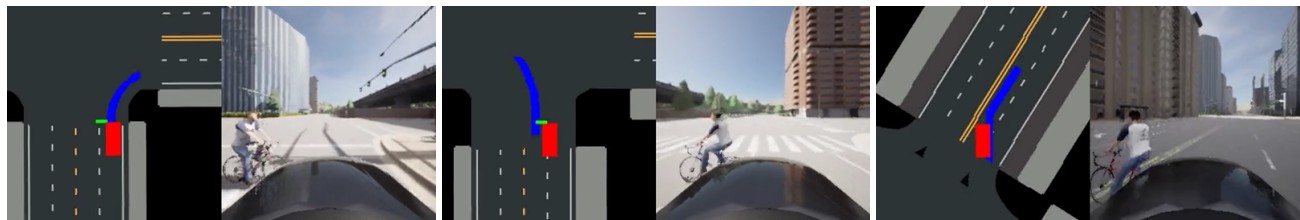

Turning Obstacle

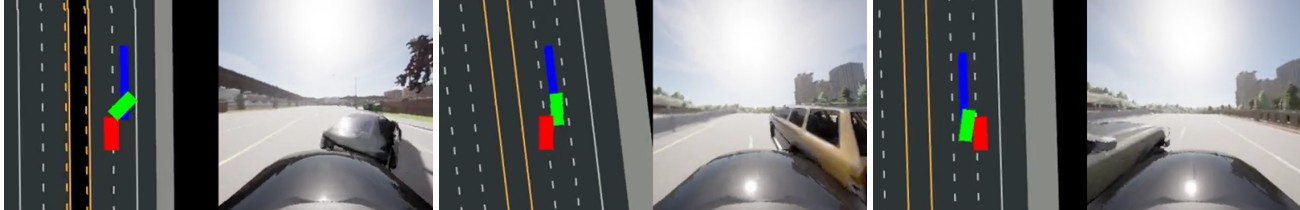

Lane Changing

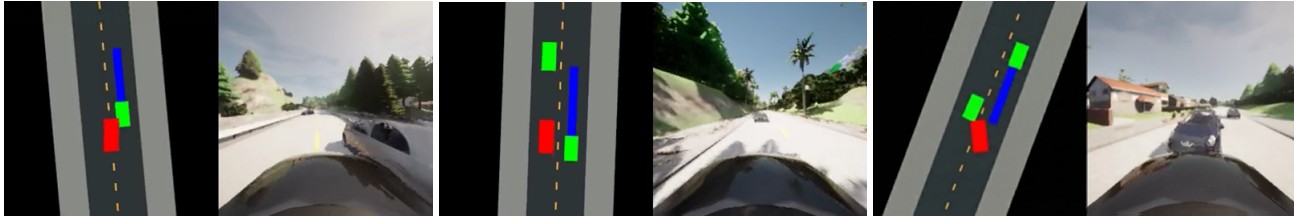

Vehicle Passing

*Table 12.* **Multi-level analysis of gernerated scenarios based on ego-vehicle behavior.** This table summarizes the average performance of different scenario generation methods, evaluated with three ego controllers over eight base scenarios. For each method, we report indicators at three levels, providing a comprehensive diagnostic of ego behavior. Safety metrics include CR (collision rate), RR (rate of running red lights), SS (rate of running stop signs), and OR (average distance traveled off road). Functionality metrics cover RF (route-following stability), Comp (fraction of route completed), and TS (time to complete the route). Etiquette metrics comprise ACC (average acceleration), YV (average yaw velocity), and LI (lane-invasion frequency). OS denotes the overall score aggregated by SafeBench.

| Algo. | Safety Level | | | | Functionality Level | | | Etiquette Level | | | OS |
|---|---|---|---|---|---|---|---|---|---|---|---|
| | CR | RR | SS | OR | RF | Comp | TS | ACC | YV | LI | |
| LC | 0.584 | 0.326 | 0.158 | 0.032 | 0.894 | 0.731 | 0.216 | 0.211 | 0.243 | 0.112 | 0.619 |
| AS | 0.586 | 0.300 | 0.160 | 0.025 | 0.891 | 0.745 | 0.261 | 0.203 | 0.245 | 0.127 | 0.620 |
| CS | 0.676 | 0.313 | 0.161 | 0.036 | 0.890 | 0.741 | 0.244 | 0.215 | 0.243 | 0.131 | 0.573 |
| AT | 0.627 | 0.312 | 0.158 | 0.028 | 0.893 | 0.726 | 0.279 | 0.219 | 0.248 | 0.137 | 0.596 |
| ChatScene | 0.831 | 0.179 | 0.143 | 0.035 | 0.833 | 0.544 | 0.223 | 0.705 | 0.532 | 0.243 | 0.482 |
| ScenePilot | **0.893** | 0.213 | 0.123 | 0.013 | 0.885 | 0.570 | 0.186 | 0.227 | 0.227 | 0.079 | **0.476** |

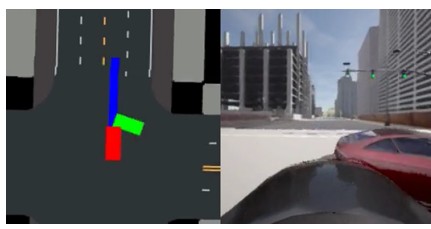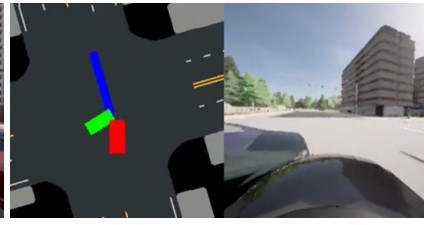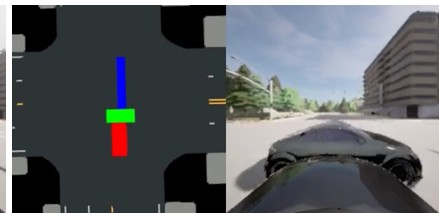

Red-light Running

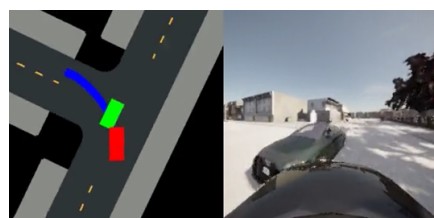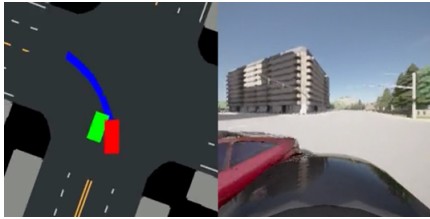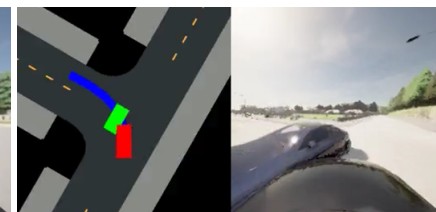

Unprotected Left-turn

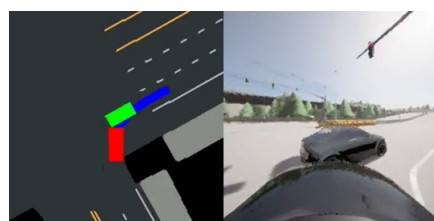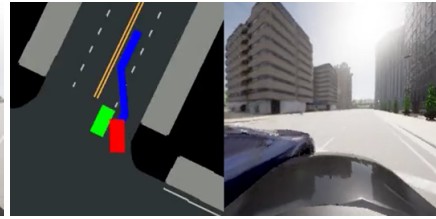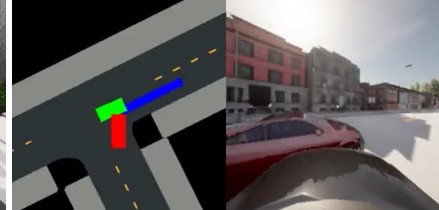

Right-turn

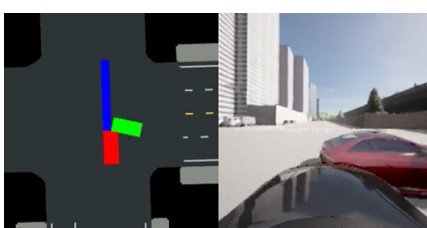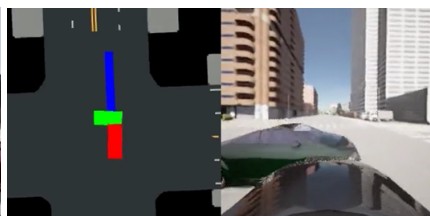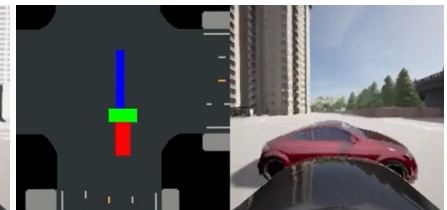

Crossing Negotiation

