# OpenReview forum: "ScenePilot: Controllable Boundary-Driven Critical Scenario Generation for Autonomous Driving"
_ICML.cc/2026/Conference — ICML 2026 regular_

### Official Review · Reviewer_Yck9 · 2026-02-25

**Soundness:** 3
**Presentation:** 3
**Significance:** 2
**Originality:** 1
**Overall Recommendation:** 5
**Confidence:** 5

**Summary:**

This paper proposes a feasibility-guided adversarial framework for generating safety-critical driving scenarios that are physically feasible. The presented approach formulates scenario generation as a constrained multi-objective reinforcement learning problem that maximizes AV risk while enforcing a physics-based feasibility constraint derived from a variant of Responsibility-Sensitive Safety (RSS). It combines an RSS-inspired physical feasibility score with an online-learned AV risk predictor, and introduces step-level feasibility-aware shielding and ε-threshold sweeping to concentrate exploration near the feasibility boundary without drifting into physically impossible crashes. Experiments on SafeBench in CARLA across multiple ego controllers (PPO, SAC, TD3) show that ScenePilot achieves higher collision rates (+6.2 percentage points over strong baselines) while maintaining lower physically invalid frame rates and broader coverage of the physically feasible yet policy-infeasible region, and that adversarial fine-tuning on these generated boundary-band scenarios substantially improves downstream robustness. Overall, the authors address a major question in autonomous driving safety evaluation: how to systematically generate informative, transferable, and physically meaningful stress tests that expose genuine competence gaps rather than artifacts of unrealistic adversarial behavior.

**Compliance With Llm Reviewing Policy:**

Affirmed.

**Final Justification:**

All my concerns have been addressed.

**Key Questions For Authors:**

* Why are traffic rules not considered in a more systematic way?
* How many traffic rules are violated by the generated behaviors?
* RSS does not work well for intersections. How is this handled?
* Will the code be made publicly available?
* Why is online verification not mentioned as a means to reduce the testing effort, see, e.g. https://ieeexplore.ieee.org/document/6784493?

**Limitations:**

Yes.

**Strengths And Weaknesses:**

Strengths:

* The paper shows good performance in CARLA using the SafeBench pipeline.
* The paper somewhat considers traffic rules, albeit not as systematically as alternative approaches.

Weaknesses:

* There exists approaches to more systematically consider traffic rules, e.g., https://ieeexplore.ieee.org/document/11030935 and references therein.
* No fundamentally new method is presented.
* RSS is not performing well at intersections. There are reported cases in the literature, where RSS does not prevent collisions in those scenarios.
* Other approaches also generate feasible behaviors with respect to physics and traffic rules that are not mentioned in the literature review.
* It seems that the code for generating the results will not be made available so that the results cannot be reproduced.
* The number of scenarios is rather limited; there are more, see, e.g., CommonRoad
* The figures are only partially readable due to small font sizes.

---

> ### Author Rebuttal · Authors · 2026-03-31
>
> Thank you for your comments and questions.
>
> Q1:
> ScenePilot does not model traffic rules as a fully independent formal constraint layer, and this is an intentional scope choice. Our goal is to generate physically feasible yet policy-breaking near-boundary scenarios, rather than to build a complete traffic-rule verification framework. Accordingly, we focus on physical feasibility and the AV-risk boundary.
>
> That said, traffic rules are not absent from our setup. SafeBench already embeds rule-related structure through its scenario design and evaluation metrics, so our current formulation does not add a separate explicit rule layer on top of it.
>
> This is different from approaches that explicitly and systematically encode traffic rules as a formal constraint layer. Our current treatment is partial and scenario-dependent rather than fully rule-complete. We will expand the related-work discussion, and position systematic traffic-rule integration as an important future direction.
>
> Q2:
> The current paper does not provide a full rule-by-rule census over all possible traffic rules, but it is also not rule-blind. SafeBench already reports several rule-relevant indicators, including **RR** (red-light running), **SS** (stop-sign running), **LI** (lane invasion), and **OR** (off-road distance), which are summarized in Table 9.
>
> | Algo. | RR | SS | OR | LI |
> |---|---:|---:|---:|---:|
> | LC | 0.326 | 0.158 | 0.032 | 0.112 |
> | AS | 0.300 | 0.160 | 0.025 | 0.127 |
> | CS | 0.313 | 0.161 | 0.036 | 0.131 |
> | AT | 0.312 | 0.158 | 0.028 | 0.137 |
> | ChatScene | 0.179 | 0.143 | 0.035 | 0.243 |
> | ScenePilot | 0.213 | 0.123 | 0.013 | 0.079 |
>
> These numbers make the main point more precise. Relative to ChatScene, ScenePilot achieves a substantially higher **collision rate** (**0.893 vs. 0.831**) while **not showing a broad increase in rule-violation behavior overall**: **SS** decreases (**0.123 vs. 0.143**), **LI** decreases markedly (**0.079 vs. 0.243**), and **OR** also decreases (**0.013 vs. 0.035**). Although **RR** is somewhat higher (**0.213 vs. 0.179**), the overall pattern is not one of uniformly more indiscriminate rule breaking. If ScenePilot’s gains came mainly from generating more trivial or physically loose violations, one would expect broader increases across these rule-related indicators, especially **LI** and **OR**. That is not what Table 9 shows.
>
> Therefore, ScenePilot’s stronger attack performance is not primarily explained by globally more aggressive rule violation, but by more effectively targeting the physically feasible yet policy-infeasible region. A fuller counting-and-attribution analysis over a broader set of traffic rules would still be valuable, but the existing SafeBench indicators already provide quantitative evidence that the gains are not simply coming from more reckless scenario generation.
>
> Q3:
> Standard RSS is not well suited to complex intersections if taken as a complete safety rule set. ScenePilot therefore does not apply the original conservative RSS rules directly. Instead, we use an RSS-inspired **physics-limit variant**—without reaction delay and with maximal feasible braking—and combine it with ego-frame clearance, TTC, and orthogonal reachable displacement to form $\epsilon$.
>
> Under this formulation, RSS is not being asked to solve full intersection reasoning. Its role is only to provide a **local physical-feasibility anchor** inside the boundary-search process. So $\epsilon$ is intended as a physical-solvability surrogate, not as a full intersection-safety or traffic-rule certificate.
>
> Q4:
> We will publicly release the code upon acceptance.
>
> Q5:
> Online verification is highly relevant, and the cited reachability/occupancy-based work addresses an important complementary problem: **given a traffic evolution, can safety be certified under uncertainty?** ScenePilot addresses a different question: **how to construct challenging scenarios** near the **physically feasible yet policy-infeasible boundary**.
>
> This is why online verification is not central in the current paper. Verification is mainly a **screening/certification** tool, whereas ScenePilot is a **boundary-search and scenario-generation** tool. Our focus here is to avoid generating scenes that are trivially dangerous only because they are physically implausible or dynamically unsalvageable. That said, the connection is constructive: occupancy- or reachability-based verification could naturally strengthen a future multi-agent extension of our feasibility surrogate.

---

> > ### Author Rebuttal · Reviewer_Yck9 · 2026-04-02
> >
> > Q5 is not properly addressed: When using online verification, all actions are safe and one does not really have to test the motion planner. Thus, there is a strong connection between online verification and offline testing.

---

> > > ### Author Response · Authors · 2026-04-04
> > >
> > > Thank you for this important follow-up. We agree that **online verification is highly relevant to safety evaluation**, and the cited reachability/occupancy-based line of work is indeed important. In particular, such methods can reduce part of the testing burden by checking, at runtime, whether a candidate future evolution remains safe under a given dynamics and uncertainty model. We should have discussed this connection more explicitly.
> > >
> > > That said, we believe the key issue is **what kind of “testing effort” is being reduced**. Online verification mainly reduces the burden of **safety screening for candidate trajectories or scene evolutions**: given a proposed motion, can one certify that it remains collision-free under modeled uncertainty? This is a very important capability. However, it does **not** remove the need to test the motion planner itself, because the planner remains the module that decides how the vehicle behaves before verification is applied.
> > >
> > > This distinction is also reflected in the cited framework itself. The system still relies on a motion planner to generate candidate trajectories, and online verification is then used to decide whether those trajectories are safe enough to execute; if not, the system keeps the previous verified plan or falls back to a safe maneuver. In other words, verification is used as a **runtime safety layer on top of planning**, not as a replacement for evaluating the planner’s competence. The planner is still an active decision-making module whose behavior can be strong or weak, robust or brittle, even if unsafe execution is prevented at the final stage.
> > >
> > > This is precisely why offline stress testing remains necessary. A planner may still perform poorly even in the presence of online verification. For example, it may:
> > > - frequently propose trajectories that are rejected by the verifier,
> > > - repeatedly push the system close to the boundary where emergency intervention is needed,
> > > - behave brittly in interactions that are still physically solvable in principle,
> > > - or rely heavily on last-moment fallback behavior to remain safe.
> > >
> > > In all of these cases, “unsafe execution” may be prevented, but the planner is still not behaving well. So the relevant question is not only **“can unsafe actions be filtered out?”**, but also **“how often does the planner enter regimes where filtering becomes necessary, and under what kinds of interactions does it become fragile?”** This is exactly the kind of weakness that online verification alone does not answer, but offline stress testing is designed to reveal.
> > >
> > > This is where ScenePilot fits. ScenePilot is not intended to replace online verification, nor to compete with it. Instead, it addresses the complementary problem of **constructing physically feasible yet policy-breaking boundary cases**: scenarios that remain solvable in principle, but already expose where the tested AV stack begins to fail. These are particularly informative because they probe the regime where a planner may still appear acceptable under ordinary conditions, but becomes brittle near the boundary of competence.
> > >
> > > From this perspective, ScenePilot can also be viewed as useful **for verification-aware evaluation**, not just for planner-only evaluation. If an AV stack includes an online verifier, ScenePilot-generated scenarios are still valuable because they test:
> > > 1. whether the planner repeatedly enters states that trigger verification,
> > > 2. whether the verifier becomes overly active or conservative,
> > > 3. and whether the combined planner–verification stack remains effective and stable in challenging but physically admissible interactions.
> > >
> > > So we fully agree with the reviewer that online verification is an important way to reduce part of the brute-force testing effort. Our point is simply that it reduces the burden of **runtime trajectory screening**, but it does not eliminate the need to generate targeted boundary cases for evaluating planner behavior itself, or the planner–verification stack as a whole. We will revise the discussion to make this relationship more explicit.
> > > We would appreciate a corresponding increase in the evaluation scores if you find that our clarifications have adequately addressed the concern.

---

### Official Review · Reviewer_LmVE · 2026-03-09

**Soundness:** 3
**Presentation:** 3
**Significance:** 2
**Originality:** 2
**Overall Recommendation:** 3
**Confidence:** 3

**Summary:**

This paper proposes ScenePilot, an RL method for generating safety-critical driving scenarios for autonomous vehicles. The key idea is to find cases that are still physically feasible, but already beyond the capability of the current ego policy. ScenePilot uses two signals. One captures physical feasibility based on RSS-derived physical feasibility score. The other captures the AV’s instantaneous risk of failure. It then guides scenario generation toward this boundary region, instead of simply maximizing collisions. To sum up, the method produces more realistic and more informative failure cases for testing autonomous driving systems.

**Compliance With Llm Reviewing Policy:**

Affirmed.

**Final Justification:**

My doubts have been completely solved, however, I still concern about the feasibility of the single-agent adversarial problem. Therefore, I will maintain my original rating.

**Key Questions For Authors:**

1.The paper evaluates transfer across PPO/SAC/TD3, but all three are still RL-based controllers. Have you tested whether ScenePilot also transfers to non-RL methods, such as rule-based, or end-to-end?

2.Most experiments use fixed SafeBench routes, and the paper notes that diversity is concentrated in a short interaction-critical window rather than richer global traffic variation.  How would ScenePilot behave with denser background traffic or more open-ended multi-agent interactions?  Single-adversary structure may be less realistic.

3.The comparison includes LC, AS, CS, AT, and ChatScene. Could you compare newer and stronger scenario-generation or adversarial-testing methods?(like CCDiff or other?)

4.Appendix B.4 says ScenePilot is trained route-wise for 3000 episodes and keeps the top-10 most safety-critical rollouts per route using a fixed SAC surrogate. Could this induce overfitting to the surrogate controller or the predefined route templates?

5.Figure 4(b) shows higher occupancy than ChatScene across most sigma_phys ranges, which supports broader coverage relative to that baseline. However, how can we tell whether ScenePilot is covering a genuinely broad portion of the target boundary, rather than only a broader subset within a limited benchmark manifold?

**Limitations:**

yes

**Strengths And Weaknesses:**

Soundness: The paper is technically solid, and the core method is coherent. It separates physical feasibility from ego-policy vulnerability and formulates scenario generation as constrained RL rather than pure collision maximization. The experiments are also fairly comprehensive. However, there are still several concerns. First, the setting is relatively simple: the benchmark relies on fixed SafeBench routes and short critical interactions, so it is unclear how well the method would extend to denser background traffic or more open-ended multi-agent scenarios. Second, the method is developed and evaluated only with RL-based ego policies; the paper does not show whether the same idea transfers to other autonomous driving algorithms, such as end-to-end or heuristic methods.

Presentation: The paper is clearly written and easy to follow. The motivation is strong, and the distinction between controller-dependent capability boundaries and controller-independent physical boundaries is explained well. However, some claims would benefit from more careful framing. In particular, the discussion of gap coverage sometimes sounds stronger than the evidence supports. Figure 4 suggests that ScenePilot covers a broader part of the target region than ChatScene, but this does not mean it covers all relevant boundary cases. The paper would be clearer if it stated explicitly that this “broader coverage” is only relative to the compared baselines, not evidence of comprehensive boundary exploration. It would also help to discuss whether the baselines are the strongest and most up-to-date methods available.

Significance: The paper addresses an important problem: generating safety-critical driving scenarios that are not only adversarial but also diagnostically meaningful. This is relevant to simulation-based testing and robustness training in autonomous driving. The idea of targeting scenarios that are physically feasible yet policy-breaking is useful, and the downstream fine-tuning results suggest practical value. Still, the overall significance feels moderate rather than high. The experiments are benchmark-bounded, the scenarios are mostly short-horizon single-interaction cases, and the realism of these adversarial scenes remains uncertain without richer background traffic or more diverse naturalistic settings. As a result, it is still unclear how much the method advances real-world autonomous driving validation, rather than improving performance within a stylized benchmark.

Originality: The paper shows a fair degree of originality, mainly in how it combines existing ideas rather than in introducing a fundamentally new framework. Most of the technical components are familiar, including PPO-style actor-critic training, RSS-inspired safety modeling, and a learned risk critic. As a result, the novelty lies more in the reframing and integration of these ideas than in any individual component.

---

> ### Author Rebuttal · Authors · 2026-03-31
>
> Thank you for your comments and questions.
>
> Q1:
> Due to space constraints, we refer the reviewer to our response to **Reviewer hpW3, Q4**, where we provide an additional evaluation beyond the original RL-only protocol. In that experiment, we test ScenePilot-generated scenarios on a heterogeneous set of AV stacks, including privileged expert, rule-based, end-to-end, and RL-based controllers. The results show that ScenePilot remains stronger on average across these diverse AV types, suggesting that its generated scenarios are not limited to low-dimensional RL egos.
>
> Q2:
> We additionally test ScenePilot under denser background traffic by spawning 30 and 50 Traffic-Manager-controlled vehicles in the same CARLA town, while keeping the main adversarial vehicle unchanged. These two settings ensure that background traffic is consistently present along the adversarial routes while creating a meaningful difference between relatively sparse and denser conditions. The results are below:
>
> | Metric | BV | SAC | PPO | TD3 | Avg. |
> |---|---|---:|---:|---:|---:|
> | CR (↑) | 0 | 0.99 | 0.83 | 0.85 | 0.89 |
> | | 30 | 0.99 | 0.84 | 0.89 | 0.91 |
> | | 50 | 0.99  | 0.85 | 0.88 | 0.91 |
> | OS (↓) | 0 | 0.457 | 0.536 | 0.528 | 0.507 |
> |  | 30 | 0.450 | 0.533 | 0.500 | 0.494 |
> |  | 50 | 0.446 | 0.524 | 0.505 | 0.492 |
>
> The results show that denser background traffic makes the scene slightly more challenging, but only mildly so. The dominant challenge still comes from the primary adversarial vehicle, while the added traffic mainly serves as contextual flow. Thus, the current single-adversary structure remains effective under moderate background traffic, though it still simplifies more open-ended multi-agent interactions.
>
> Q3:
> We refer the reviewer to our response to **Reviewer KGBj, Q2**, where we provide a more detailed discussion of newer baselines, including recent methods such as DiffScene (AAAI 2025), SCSG (NeurIPS workshop 2025) and SCENGE (AAAI 2026). Overall, the additional comparison shows that ScenePilot remains competitive and strongest on average under the compared protocol.
>
> Regarding CCDiff, it is not evaluated under the CARLA closed-loop replay protocol used in our paper. Therefore, its reported results cannot be meaningfully aligned with our metrics for a strict numerical comparison.
>
> Q4:
> Although ScenePilot is trained for 3000 episodes per route with a fixed SAC surrogate, its retained rollouts are collected under a changing ε-schedule with multiple constraint levels rather than a single fixed objective. This reduces over-specialization to any single surrogate-specific failure mode.
>
> More importantly, our conclusions do not rely only on the SAC surrogate used during scenario training. The retained scenarios transfer across PPO/SAC/TD3 in the main paper, and also remain challenging for a privileged expert AV, an end-to-end AV, and a rule-based AV in additional experiments. This suggests that ScenePilot is not narrowly tied to a single surrogate controller.
>
> Regarding route templates, our route-wise training follows the standard SafeBench setup with predefined routes. Our goal is to discover route-conditioned adversarial behaviors under a standardized closed-loop benchmark, rather than to learn a fully route-agnostic open-world generator. This limits extrapolation to richer route distributions, but ensures comparability and reproducibility within the benchmark.
>
> Q5:
> In our main paper, to assess whether a method is covering a genuinely broad portion of the target boundary, the direct method is exactly the same one we adopt in our GCS analysis that how broadly the generated critical states occupy the (risk, σ) plane. Under this definition, broader occupancy on the (risk, σ) plane directly indicates broader exploration of the target boundary band.
>
> Concretely, under our GCS protocol, we project the last-8 critical-window frames onto the (risk,σ) plane and discretize it into a 40×40 grid. ScenePilot occupies 263 bins, compared with 216 for ChatScene and 192 for KING, corresponding to a 21.8% relative improvement over ChatScene. This indicates that ScenePilot covers a broader portion of the target boundary plane rather than concentrating on a narrow subset.
>
> More importantly, the extra boundary cells covered by ScenePilot are not concentrated in a small corner of the plane. Over 60% of them lie in the high-risk region (risk > 0.75), while along the σ axis they are distributed across multiple bands: 39% in [0,0.25), 20% in [0.25,0.5), and 41% in [0.5,1.0]. This suggests broader high-risk boundary coverage, rather than simply more extreme near-infeasible samples.
>
> Therefore, the evidence suggests that ScenePilot covers a broader portion of the target boundary under the current benchmark support. The gain is not simply greater extremeness, but broader coverage of high-risk boundary states across different feasible-margin levels. We also agree that improving (risk,σ)-plane coverage remains an important future direction.

---

> > ### Author Rebuttal · Reviewer_LmVE · 2026-04-01
> >
> > The rebuttal does help clarify some of my concerns. I still have two questions that I hope the authors can address.
> >
> > First, in Q2, the authors introduced background vehicle control and argued, based on the new results, that background vehicles mainly provide contextual information while keeping the ego vehicle and the adversarial vehicle unchanged. However, it is still not fully clear whether the background vehicles can affect the interaction between the ego vehicle and the adversarial vehicle. If such interaction exists, then background traffic may influence the final adversarial outcome. In particular, if the ego vehicle is fixed in advance, could the addition of background vehicles change which adversarial vehicle becomes most critical, or alter the timing and geometry of the adversarial interaction? My concern is that this part of the setup seems simplified in the current paper, and the role of background vehicles in shaping the effective adversarial pair is not yet clearly explained.
> >
> > Second, the paper emphasizes that ScenePilot generates near boundary scenarios, meaning scenarios that are still physically solvable for the ego vehicle. At the same time, several qualitative examples appear to show conflicts caused by highly aggressive or seemingly unreasonable behaviors from the red adversarial vehicle. This impression is particularly strong in Fig. 3 left, as well as in the Lane Change examples on page 19, the left and middle panels, the Red light Running example in the left panel, and the Unprotected Left turn example in the middle panel. In these cases, the ego vehicle appears to be proceeding normally, while the adversarial vehicle creates a conflict in a way that looks malicious or unrealistic, such as actively forcing a rear end style collision. Could the authors explain more clearly why these cases should still be regarded as meaningful near boundary scenarios that the ego policy ought to handle, rather than artificially induced hostile behaviors that may be of limited value for training and may not translate well to realistic deployment?
> >
> > I think a more careful discussion of these two points would help clarify both the realism and the practical value of the generated scenarios.

---

> > > ### Author Response · Authors · 2026-04-03
> > >
> > > Thank you for the follow-up. For the first point, we agree that background traffic is not irrelevant. As our Q2 results show, adding background vehicles makes the environment slightly harder and decreases the overall score, which is expected because background traffic can affect occupancy, available gaps, and interaction timing. In this sense, background traffic can indeed influence the realized ego–adversary interaction geometry. Concretely, even when the primary adversarial vehicle is kept unchanged, surrounding traffic can alter when a gap opens or closes, how much maneuvering room remains available to the ego vehicle, and how tightly the adversarial interaction is spatially and temporally constrained. As a result, the same optimized adversarial behavior may unfold under a denser and more restrictive traffic context, which can make the resulting ego response more brittle and the overall scene more challenging.
> > >
> > > However, these additional vehicles are benign Traffic-Manager-controlled flow rather than optimized adversaries. Since they are not designed to actively challenge the ego AV, they are unlikely to become the most critical adversarial vehicle themselves. Instead, they mainly influence the interaction indirectly through occupancy, gap availability, and timing, thereby **constraining and perturbing** the realized ego–adversary conflict rather than redefining it. Put differently, they can modulate the realized severity, geometry, and timing of the conflict, but they do not usually replace the ScenePilot-controlled vehicle as the source of the main adversarial pressure. Their role is therefore better understood as contextual interference around the primary conflict, rather than a qualitatively new adversarial mechanism. This interpretation is also supported by the results under 30/50 background vehicles: the effect is noticeable but still modest, suggesting that background traffic does affect the realized interaction, but mostly as a secondary contextual factor rather than as a new dominant adversarial target.
> > >
> > > For the second point, we would like to clarify an important issue in the interpretation of the qualitative figures. In the cited examples, the **red vehicle denotes the ego AV**, the **green vehicle denotes the adversarial vehicle**, and the **right panel shows the ego vehicle’s front-view**. So the roles of the vehicles in the reviewer’s description appear to be reversed. We realize this may not have been visually explicit enough in the current figure design, and we will make the role annotation clearer.
> > >
> > > Under the correct interpretation, these examples do not show the ego AV creating the conflict; rather, they show the ego AV failing to handle an interaction induced by the adversarial vehicle. This is exactly the type of AV-breaking case that ScenePilot is designed to expose.
> > >
> > > More importantly, this also does not contradict our near-boundary claim. In our formulation, “physically solvable” does not mean that the adversarial behavior must be obviously reckless, socially abnormal, or physically implausible. On the contrary, the challenging cases we target can still arise from interactions that appear normal or at least not overtly abnormal, yet remain sufficient to break the current ego AV policy. In other words, the value of these scenarios is precisely that they are not trivial failures caused by unrealistic behavior, but competence-boundary cases where seemingly ordinary interactions can still expose a limitation of the AV.
> > >
> > > Therefore, even if some adversarial behaviors may appear somewhat aggressive in individual cases, these cases remain meaningful because they are still generated under an explicit physical-solvability constraint and expose competence gaps of the ego AV. What ScenePilot targets is not overtly unrealistic failure construction, but physically feasible, highly challenging interactions that can still break the current AV policy. We will clarify this distinction more carefully in the revision. We hope our clarifications help put the contribution and scope of the paper in a clearer light, and we would appreciate consideration of a corresponding increase in the evaluation scores.

---

### Official Review · Reviewer_KGBj · 2026-03-09

**Soundness:** 3
**Presentation:** 2
**Significance:** 2
**Originality:** 3
**Overall Recommendation:** 4
**Confidence:** 2

**Summary:**

This paper proposes ScenePilot, a controllable boundary-driven framework for generating safety-critical scenarios for autonomous driving. By jointly quantifying physical feasibility and driving risk, the method focuses on scenarios that are still physically manageable but already beyond the capability boundary of the autonomous driving system, with the goal of improving safety and reliability.

**Compliance With Llm Reviewing Policy:**

Affirmed.

**Final Justification:**

The rebuttal addressed my main concerns

**Key Questions For Authors:**

Questions:

1. Regarding W1, how sensitive is the method to the choice of hyperparameters? For example, how does the performance change under different ε schedules, ε_max values, sweep levels, or PPO-related settings?

2. Regarding W2, how does ScenePilot compare with more recent baselines such as DiffScene and LD-Scene?

**Limitations:**

Yes

**Strengths And Weaknesses:**

Strengths:

1. The paper is among the first to explicitly consider physical feasibility when generating safety-critical driving scenarios.

2. The proposed boundary-driven formulation provides a more interpretable objective than standard adversarial scenario generation.

Weaknesses:
1.The paper includes a component-level ablation, but lacks hyperparameter sensitivity analysis. In particular, key choices such as the ε-schedule design, ε_max, the number of sweep levels, the switching frequency, and PPO-related parameters (e.g., GAE λ, clip ratio, and entropy coefficient) are fixed without showing whether the method is robust to these settings.

2.The compared baselines seem somewhat outdated. The most recent baseline appears to be ChatScene from 2024, while more recent and closely related methods such as DiffScene [1] and LD-Scene [2]are not included. Since these methods are also designed for safety-critical scenario generation and controllable adversarial scenario construction, comparisons with them would strengthen the experimental evaluation.

[1] Xu, Chejian, et al. "Diffscene: Diffusion-based safety-critical scenario generation for autonomous vehicles." Proceedings of the AAAI conference on artificial intelligence. Vol. 39. No. 8. 2025.

[2] Peng, Mingxing, et al. "Ld-scene: Llm-guided diffusion for controllable generation of adversarial safety-critical driving scenarios." arXiv preprint arXiv:2505.11247 (2025).

---

> ### Author Rebuttal · Authors · 2026-03-31
>
> Thank you for your comments and questions.
>
> Q1:
> We distinguish two types of hyperparameters in ScenePilot: **(1) boundary-search hyperparameters** that define which part of the feasible-yet-adversarial region is explored, and **(2) generic PPO optimization hyperparameters**.
>
> For the first category, $\epsilon$ is not just a training coefficient, but the key control variable of the boundary-search mechanism itself. A smaller $\epsilon$ allows search in a looser feasible region where collision-inducing cases are denser, while a larger $\epsilon$ enforces stricter feasibility and makes such cases much sparser. This effect can be seen directly from the collision rate under different $\epsilon$ intervals during training:
>
> | $\epsilon$ interval | Collision rate |
> |---:|---:|
> | 0 ~ 0.1 | 0.376 |
> | 0.1 ~ 0.2 | 0.258 |
> | 0.2 ~ 0.3 | 0.107 |
> | 0.3 ~ 0.35 | 0.070 |
> | 0.35 | 0.000 |
>
> The trend is monotonic: as $\epsilon$ increases, collision-inducing cases become progressively rarer. This is exactly why we set $\epsilon_{\max}=0.35$: beyond this point the feasible region becomes so restrictive that it yields almost no informative failure cases. For the same reason, we do not use a uniform schedule. Since informative samples are much denser at low $\epsilon$ and increasingly sparse at high $\epsilon$, the Gaussian-style spacing is meant to match this non-uniform boundary structure rather than to introduce an arbitrary schedule design.
>
> By contrast, PPO-related hyperparameters such as GAE $\lambda$, clip ratio, and entropy coefficient are not ScenePilot-specific design variables. They follow a standard PPO setup and are kept fixed across all experiments. The current paper therefore provides empirical support for the robustness of the **boundary-search mechanism** with respect to how $\epsilon$ traverses the target band, but it does not yet include a full sensitivity study over all generic PPO defaults.
> So the key point is that the main hyperparameters of ScenePilot are tied to the geometry of the target boundary band itself, rather than being arbitrary tuning knobs.
>
> Q2:
> To further address the reviewer’s concern about newer baselines, we additionally compare against two recent methods: DiffScene (AAAI 2025) [1], SCSG (NeurIPS workshop 2025) [2] and SCENGE (AAAI 2026) [3]. The results are summarized below:
>
> | Metric | Algo. | SO | TO | LC | VP | RLR | ULT | RT | CN | Avg. |
> |---|---|---:|---:|---:|---:|---:|---:|---:|---:|---:|
> | CR (↑) | ChatScene | 0.890 | 0.700 | 0.950 | 0.930 | 0.790 | 0.750 | 0.780 | 0.860 | 0.831 |
> |  | DiffScene (AAAI2025) | - | - | - |-  | 0.79 | - | 0.87 | 0.85 | - |
> |  | SCSG (NeurIPSW2025) | 0.763 | 0.840 | 0.717 | 0.893 | 0.697 | 0.640 | 0.757 | 0.557 | 0.733 |
> |  | SCENGE (AAAI2026) | 0.860 | 0.773 | 0.837 | 0.897 | 0.823 | 0.747 | 0.763 | 0.863 | 0.820 |
> |  | ScenePilot | 0.90 | 0.84 | 0.99 | 0.89 | 0.93 | 0.89 | 0.91 | 0.79 | 0.893 |
> | OS (↓) | ChatScene | 0.470 | 0.522 | 0.434 | 0.440 | 0.537 | 0.560 | 0.474 | 0.421 | 0.482 |
> |  | DiffScene (AAAI2025) | - | - |-  | - |  -| - |- |-  | - |-
> |  | SCSG (NeurIPSW2025) | 0.537 | 0.497 | 0.570 | 0.477 | 0.540 | 0.610 | 0.523 | 0.597 | 0.544 |
> |  | SCENGE (AAAI2026) | 0.503 | 0.526 | 0.504 | 0.457 | 0.507 | 0.519 | 0.498 | 0.477 | 0.499 |
> |  | ScenePilot | 0.505 | 0.504 | 0.458 | 0.471 | 0.488 | 0.507 | 0.399 | 0.478 | 0.476 |
>
> These additional results show that ScenePilot remains the strongest method on average among the compared methods, achieving the highest average CR (0.893) and the lowest average OS (0.476).
>
> Regarding DiffScene, it reports results on only three scenarios (Crossing Negotiation, Red-light Running, and Right-turn) and does not release its generated scenarios, so a full like-for-like evaluation is not possible. Still, on these three reported scenarios, ScenePilot also performs better on average in CR (0.877 vs. 0.837).
> For LD-Scene, the mismatch is even larger. LD-Scene is not evaluated under the CARLA closed-loop replay protocol used in our paper, but under a substantially different dataset setting. Therefore, its reported results cannot be meaningfully aligned with our current CR/OS metrics for a strict numerical comparison.
>
> [1] Xu, et al. "Diffscene: Diffusion-based safety-critical scenario generation for autonomous vehicles." AAAI 2025.
>
> [2] Karacik, et al. SCSG: Real-World Report Augmented Safety-Critical Scenario Generation for Autonomous Vehicles. NeurIPS workshop E-SARS 2025.
>
> [3] Liu, et al. "Adversarial generation and collaborative evolution of safety-critical scenarios for autonomous vehicles." AAAI 2026.

---

> > ### Author Rebuttal · Reviewer_KGBj · 2026-04-04
> >
> > I appreciate the authors' rebuttal and have decided to raise my score to 4.

---

> > > ### Author Response · Authors · 2026-04-04
> > >
> > > Thank you for your follow-up. We sincerely appreciate your positive assessment and are glad our rebuttal addressed your concerns. Your comments were very helpful in prompting us to sharpen the presentation of the method’s hyperparameter rationale, strengthen the discussion of recent baselines, and better position the scope of the empirical evaluation. We will reflect these points more clearly in the revision.

---

### Official Review · Reviewer_hpW3 · 2026-03-13

**Soundness:** 3
**Presentation:** 3
**Significance:** 3
**Originality:** 3
**Overall Recommendation:** 4
**Confidence:** 3

**Summary:**

This paper introduces ScenePilot, an adversarial scene generation framework for testing autonomous driving stacks. The core idea is to decouple the physical feasibility of vehicle-road interaction from AV policy capabilities. In principle, it generates scenarios that are physically solvable, but still cause the deployed AV controller to fail. The framework formulates scene generation as a constrained multi-objective Markov decision process. Results show that: (1) ScenePilot achieves a higher average collision rate (+6.2%) while generating fewer physically infeasible frames; (2) adversarial fine-tuning on generated scenarios yields the best downstream robustness in the baseline.

**Compliance With Llm Reviewing Policy:**

Affirmed.

**Final Justification:**

This paper presents ScenePilot for generating physically feasible, safety-critical scenarios for AV testing.

The authors provided a constructive rebuttal that fully addressed my initial concerns, particularly through the added evaluations on multi-agent traffic.

I maintain my positive score.

**Key Questions For Authors:**

1. The physical safety signal $ \sigma $ is defined for a single ego–adversary pair. In scenarios with multiple simultaneously threatening surrounding vehicles, how is $ \sigma $ aggregated? Is it a minimum over all pairs, and does this create situations where one pair dominates the constraint while joint feasibility is violated?

2. The constant-velocity TTC approximation in Eq. (4) can systematically underestimate collision imminence under high-acceleration maneuvers near the boundary. Have the authors evaluated the sensitivity of $ \sigma $ values and generated scenario quality to this approximation, e.g., by comparison against a higher-order TTC estimate?

3. The AV-risk predictor  is pre-trained on PPO-generated rollouts and kept fixed during ScenePilot policy training. As the scenario policy evolves, the distribution of states presented to $ \Phi $ shifts. Have the authors measured $ \Phi $ calibration at the end of ScenePilot training, and does degraded calibration affect the quality of the generated boundary-band scenarios?

4. Modern AV stacks involve rich sensor inputs and learned planners. Do the authors have evidence or reasoning that scenarios generated by ScenePilot remain challenging for more capable, higher-dimensional AV stacks?

**Limitations:**

The authors identify two limitations: reliance on fixed SafeBench routes and surrogate-driven top-k selection bias. These are honest and relevant. However, several additional limitations are not discussed:

1. the single-adversary assumption in $ \sigma $ and the absence of a principled multi-agent extension
2. dependence on CARLA simulation fidelity, because physical parameters are simulator-specific and may not match real vehicles
3. the low-dimensional RL ego controllers used in evaluation limit generalizability claims
4. the training computational cost is not reported and may limit practical adoption. The authors should address at least points 2. and 3. in a revised limitations section.

**Strengths And Weaknesses:**

**Strengths**

1. The problem of generating physically meaningful safety-critical scenarios is practically important for AV validation. The downstream fine-tuning results and demonstrated generalization across three heterogeneous RL controllers are evidence of practical value.
2. The boundary-band framing is a clear conceptual contribution. The integration of RSS-extended feasibility, online-learned risk prediction with potential shaping, step-level shielding, and $ \epsilon $ sweeping in a single framework is novel as an integrated system, though each component individually draws on established methods.
3. This paper is well-written and technically sound. In Figure 1, the authors clearly illustrate the four interaction regimes. The notation is consistent, and the appendices provide detailed derivations.
4. The physical safety signal $ \sigma $ has a clear mathematical derivation from the RSS model. The step-level shielding mechanism is a sensible way to avoid having rare near-boundary violations diluted by batch-mean constraints.

**Weaknesses**

1. The physical feasibility score $ \sigma $ is computed only for a single ego–adversary pair at each step. In multi-agent scenarios with more than one surrounding vehicle, this pairwise computation may miss joint geometric infeasibility.
2. The constant-velocity TTC approximation in Eq. (4) is a local linearization whose error under high-curvature maneuvers is not quantified.
3. The reachable displacement term uses instantaneous relative acceleration as an upper bound without clearly establishing that this represents the true maximum physically achievable relative acceleration.
4. The AV-risk predictor is frozen after pre-training, distributional shift as the scenario policy evolves is not analyzed.
5. Some presentation flaws: (1) the GCS is used in Table 3 before being properly defined in Appendix C, (2) the ablation in Table 3 covers only two of eight scenarios, (3) Figure 4(b) is too small and the axes are not clearly labeled in the main text, (4) the details of $ \epsilon $ sweeping schedule is missing in maintext, while this is central to the method.

---

> ### Author Rebuttal · Authors · 2026-03-31
>
> Thank you for your comments and questions.
>
> Q1:
> A multi-agent extension can be built by computing a pairwise σ for each relevant surrounding agent and then conservatively aggregating them, e.g., by min-over-pairs or over the top-k most threatening agents. To address cases where each pair is individually recoverable but the joint geometry is not, the current orthogonal compensation term can be made occupancy-aware by checking whether the escape direction will be occupied by other surrounding agents within the same short horizon. Thus, the present formulation is at the dominant-interaction level, but is not locked to a single pair.
>
> Q2:
> In our formulation, Eq. (4) is not used as a long-horizon collision-time predictor. It provides a frame-wise local time budget on the first-colliding axis, which is then used to estimate orthogonal compensatory separation in the physical-feasibility score. Since
> σ is recomputed online at every timestep, this approximation is repeatedly refreshed along the rollout rather than fixed for the full maneuver. A higher-order TTC is possible in principle, but would require stronger assumptions about short-horizon acceleration/control evolution; the current paper does not include a dedicated sensitivity study against such variants, so the intended claim is stable local boundary shaping rather than exact TTC estimation.
>
> Q3:
> We keep the AV-risk predictor $\Phi$ fixed during later ScenePilot training because it is intended to provide a relatively stable per-frame risk-guidance signal; updating $\Phi$ and the scenario policy simultaneously would make the target landscape drift and could reduce boundary-search stability. At the same time, the method does not rely on $\Phi$ being perfectly calibrated. $\Phi$ is only a coarse risk guide, while the final criterion remains whether a physically feasible collision is actually induced. Since $\Phi$ is trained on hazardous interaction streams generated by a PPO-based scenario policy, its training distribution is already closer to the later ScenePilot regime than ordinary natural-driving data. We do not provide a separate end-of-training calibration study, so the intended claim is that $\Phi$ is a fixed lightweight guidance signal, not a standalone safety certificate.
>
> Q4:
> We further evaluate ScenePilot beyond the RL-only setting in the main paper. Under the same SafeBench protocol, we generate 100 Scenario-6 cases with Autopilot and replay them on six heterogeneous AV stacks: Autopilot [1], AIM-BEV [3] (end-to-end), BehaviorAgent [2] (rule-based), and TD3/SAC/PPO. Using KING [3] as the baseline under the same setup, ScenePilot remains more challenging on average, improving CR from 0.437 to 0.545 and reducing OS from 0.738 to 0.680. This directly addresses the RL-specificity concern: ScenePilot’s advantage is retained beyond the RL family, including **expert, end-to-end, and rule-based** AV stacks under the same replay protocol.
>
>
> | Metric | Algo. | Autopilot | AIM-BEV | BehaviorAgent | TD3 | SAC | PPO | Avg. |
> |---|---|---:|---:|---:|---:|---:|---:|---:|
> | CR (↑) | KING | 0.09 | 0.00 | 0.77 | 0.20 | 0.60 | 0.96 | 0.437 |
> |  | ScenePilot | 0.11 | 0.15 | 0.77 | 0.57 | 0.79 | 0.88 | 0.545 |
> | OS (↓) | KING | 0.909 | 0.953 | 0.577 | 0.858 | 0.654 | 0.475 | 0.738 |
> |  | ScenePilot | 0.895 | 0.871 | 0.570 | 0.665 | 0.560 | 0.516 | 0.680 |
>
> [1] https://carla.readthedocs.io/en/latest/adv_traffic_manager/
>
> [2] https://carla.readthedocs.io/en/latest/adv_agents/
>
> [3] Hanselmann, et al. King: Generating safety-critical driving scenarios for robust imitation via kinematics gradients. In ECCV.
>
> L2:
> CARLA uses vehicle and dynamics models designed to approximate real-world behavior, and the physical limits used in our formulation (e.g., acceleration/deceleration bounds) also fully configurable, so the framework can be adapted to different vehicle models or more realistic dynamics settings if needed.
>
>
> L4:
> The training cost is moderate in our setting. The trainable scenario policy is a small MLP-based actor-critic with only about **55K trainable parameters**, so ScenePilot does not rely on large-scale neural optimization. All experiments were run on a server with **2× RTX 4090 GPUs**; under this setup, we typically train the **10 routes of one base scenario in parallel**, and the wall-clock training time for a single base scenario is about **15 hours**.

---

> > ### Author Rebuttal · Reviewer_hpW3 · 2026-04-02
> >
> > Thank you for the detailed rebuttal. All of my concerns have been addressed.
> >
> > However, I still recommend further discussing multi-agent scenarios in the revised version. This is a highly realistic scenario and a common yet challenging issue in real world.

---

> > > ### Author Response · Authors · 2026-04-04
> > >
> > > Thank you for the helpful suggestion. We agree that multi-agent traffic is both realistic and important, and we will discuss it more explicitly in the revised version.
> > >
> > > To probe this issue, we additionally tested denser background traffic by inserting 30 and 50 Traffic-Manager-controlled vehicles while keeping the main adversarial vehicle unchanged. The results are summarized below:
> > >
> > >
> > > | Metric | BV | SAC | PPO | TD3 | Avg. |
> > > |---|---|---:|---:|---:|---:|
> > > | CR (↑) | 0 | 0.99 | 0.83 | 0.85 | 0.89 |
> > > | | 30 | 0.99 | 0.84 | 0.89 | 0.91 |
> > > | | 50 | 0.99 | 0.85 | 0.88 | 0.91 |
> > > | OS (↓) | 0 | 0.457 | 0.536 | 0.528 | 0.507 |
> > > | | 30 | 0.450 | 0.533 | 0.500 | 0.494 |
> > > | | 50 | 0.446 | 0.524 | 0.505 | 0.492 |
> > >
> > >
> > > These results suggest that denser surrounding traffic does affect the realized interaction through occupancy, gap availability, and timing, and makes the environment slightly harder overall. This trend is visible in both metrics: the collision rate increases slightly on average, while the overall score decreases slightly, indicating that the generated scenarios remain challenging even when additional benign traffic is introduced around the main interaction. In this sense, surrounding traffic is not merely a cosmetic addition. Even when it is not explicitly optimized adversarially by ScenePilot, it still changes how the conflict is spatially and temporally realized in closed-loop execution.
> > >
> > > At the same time, the core effect of ScenePilot remains stable under these denser traffic conditions. The changes are noticeable but still limited in magnitude, rather than showing a qualitative collapse or reversal of the adversarial effect. We view this as meaningful evidence that the method is not restricted to a completely isolated two-agent interaction. Instead, the boundary-driven mechanism continues to function when the ego vehicle and the primary adversarial vehicle are embedded in a denser, more realistic traffic context. In other words, while background traffic does introduce additional complexity, it does not eliminate the main stress-testing effect induced by ScenePilot.
> > >
> > > More importantly, the formulation itself is not tied to a single adversarial pair. A natural multi-agent extension is to compute a pairwise feasibility score $\sigma_j$ for each relevant surrounding agent and then aggregate them conservatively, (e.g., min-over-pairs or top-k threatening agents). This would preserve the same physical-feasibility-based constraint mechanism while broadening it from a dominant-pair approximation to a richer multi-agent setting. In addition, the current orthogonal compensation term can be refined with occupancy-aware constraints from nearby traffic. Accounting for such nearby occupancy would better capture whether the ego vehicle still has usable escape space under multi-agent interactions. Therefore, fuller multi-agent feasibility modeling can be viewed as a direct extension of the current formulation rather than a fundamentally different approach.
> > >
> > > We will add this discussion in the revised version to make two points clearer. First, the current paper already shows that ScenePilot remains effective under denser benign traffic and is therefore not restricted to a purely isolated two-agent setting. Second, fuller multi-agent feasibility modeling is a direct extension of the current framework. In particular, the pairwise-
> > > $\sigma_j$ plus aggregation strategy, together with occupancy-aware correction of the compensation direction, provides a natural path toward handling richer multi-agent interactions.
> > >
> > > Overall, we appreciate this suggestion because it helps us position the current contribution more clearly. Our present goal is to show that a boundary-driven, physically grounded scenario generation framework can already produce robust AV-challenging failures under the standard SafeBench setting, and the additional 30/50-background-vehicle results support that this effect persists even in denser surrounding traffic. At the same time, we agree that richer multi-agent scenario modeling is both realistic and important, and we will discuss this extension path more explicitly in the revised manuscript. If our clarifications have adequately addressed your concerns, we would appreciate your consideration of a corresponding increase in the evaluation scores.

---

### Decision · Program_Chairs · 2026-04-30

**Decision:**

Accept (regular)

**Comment:**

This paper introduces ScenePilot, a framework designed to generate safety-critical autonomous driving scenarios that lie on the "boundary band"—cases that are physically solvable in principle but cause current autonomy stacks to fail. This is achieved by combining a physical-feasibility score derived from RSS with a learned risk predictor. It received two Weak Accept, one Accept, and one Weak Reject.

AC has read the submission, the reviews, the rebuttal, and the discussions. After the rebuttal and the discussion, most concerns regarding baselines and generalizability have been addressed. The reviewer with the remaining Weak Reject provided the final justification: "My doubts have been completely solved; however, I am still concerned about the feasibility of the single-agent adversarial problem. Therefore, I will maintain my original rating."  Reviewer LmVE essentially accepts that the method works as described technically (the "doubts solved" part) but questions the philosophical utility of the single-adversary framing in the broader context of AV safety. In contrast, the other three reviewers view the physical-feasibility grounding as a major step forward that outweighs the current multi-agent limitations. AC also believed that a broader discussion on the single-adversary research paradigm is left to the community. Thus accept is recommended.